# Predicted deleterious mutations reveal the genetic architecture of male reproductive success in a lekking bird

R. S. Chen [1] ✉, C. D. Soulsbury [2], K. Hench [1], K. van Oers [3,4] & J. I. Hoffman [1,5,6,7]

Deleterious mutations are ubiquitous in natural populations and, when expressed, reduce fitness. However, the specific nature of these mutations and the ways in which they impact fitness remain poorly understood. We exploited recent advances in genomics to predict deleterious mutations in the black grouse (*Lyrurus tetrix*), an iconic lekking species. Analysis of 190 whole genomes alongside comprehensive life-history data including repeated measures of behavioural, ornamental and fitness traits revealed that deleterious mutations identified through evolutionary conservation and functional prediction are associated with reduced male lifetime mating success. Both homozygous and heterozygous deleterious mutations reduce fitness, indicating that fully and partially recessive mutations contribute towards an individual's realized mutation load. Notably, deleterious mutations in promotors have disproportionally negative fitness effects, suggesting that they impair an individual's ability to dynamically adjust gene expression to meet context-dependent functional demands. Finally, deleterious mutations impact male mating success by reducing lek attendance rather than by altering the expression of ornamental traits, suggesting that behaviour serves as an honest indicator of genetic quality. These findings offer insights into the genetic architecture of male fitness and illuminate the complex interplay between genetic variation and phenotypic expression.

Deleterious mutations segregate in all natural populations, reducing fitness when expressed and contributing to an individual's mutation load (the reduction in fitness due to the accumulation of deleterious genetic variants[1]). Theory on the fitness effects of deleterious mutations is well-established[1–3] and empirical work has shown that induced mutations (for example, through ionizing radiation) can disrupt sexual trait expression[4,5] and reduce fitness[6–8]. However, key attributes of the deleterious mutations contributing to the mutation load remain poorly understood, including their effect sizes and dominance coefficients, whether they are located in coding or non-coding regions of the genome, and how they influence reproductive success via the expression of sexually selected traits[9]. Addressing these knowledge gaps is essential for understanding the evolutionary dynamics of the mutation load and the genetic architecture of fitness.

Recent advances in whole genome resequencing and bioinformatics now allow for the prediction of deleterious mutations from

[1]Department of Evolutionary Population Genetics, Faculty of Biology, Bielefeld University, Bielefeld, Germany. [2]School of Natural Sciences, Joseph Banks Laboratories, University of Lincoln, Lincoln, UK. [3]Department of Animal Ecology, Netherlands Institute of Ecology (NIOO-KNAW), Wageningen, the Netherlands. [4]Behavioural Ecology Group, Wageningen University & Research (WUR), Wageningen, the Netherlands. [5]Center for Biotechnology (CeBiTec), Faculty of Biology, Bielefeld University, Bielefeld, Germany. [6]British Antarctic Survey, Cambridge, UK. [7]Joint Institute for Individualisation in a Changing Environment (JICE), Bielefeld University and University of Münster, Bielefeld, Germany. ✉e-mail: rebecca.chen@uni-bielefeld.de

genomic data, even in non-model organisms[1,10]. Two prediction approaches are commonly used: evolutionary conservation, which assumes that mutations in conserved regions are detrimental[11], and functional prediction, which evaluates the potential impact of mutations on protein structure and function[12,13]. The resulting predicted deleterious mutations can be aggregated to estimate individual genomic mutation loads, which are often used as proxy measures of the genetic health of endangered species[14–16]. However, empirical validation of the assumptions behind these approaches remains limited, with recent studies focusing on functional predictions and using modest sample sizes[17,18].

Studies of inbreeding depression typically infer the fitness effects of deleterious mutations indirectly by assuming that the expression of recessive deleterious alleles across the genome scales in proportion to genome-wide homozygosity[19]. However, inbreeding coefficients are not strictly informative about the number, genomic distribution and fitness effects of deleterious mutations across an individual's genome, nor do they account for the effects of heterozygous deleterious mutations. Theory suggests that partially recessive deleterious mutations expressed in the heterozygous state also reduce fitness[20] and thereby contribute to the realized load (the fraction of the total mutation load that is expressed in the current generation[1]). Accordingly, individual genomic mutation load estimates, which incorporate information on both homozygous and heterozygous variants, should theoretically be stronger predictors of fitness than genomic inbreeding coefficients. In practice, however, the actual explanatory power of these measures will depend on the precision of their estimation.

Once dismissed as 'junk DNA', non-coding regions, including regulatory elements such as promoters[21], are increasingly recognized for their functional significance[22,23]. However, the extent to which mutations in non-coding regions affect phenotypes differently from those in coding regions remains unclear[24]. Deleterious mutations in non-coding regions that disrupt gene regulation may reduce fitness by impairing an organism's ability to dynamically adjust gene expression to meet context-dependent functional demands[25]. This may be particularly relevant in the context of sexual selection as mating strategies and decisions depend on multiple factors that change over time, including age[26] and body condition[27]. Consequently, investigating the effects of deleterious mutations across different genomic regions could produce new insights into the relationship between genetic variation and fitness.

Finally, deleterious mutations may impact male lifetime reproductive success directly or indirectly by influencing the expression of sexual traits, which can serve as honest indicators of immune function[28] and body condition[29], potentially signalling genetic quality. The black grouse (*Lyrurus tetrix*) is a lekking galliform that exhibits extremely strong sexual selection, with both male–male precopulatory competition and female choice playing important roles. Sexual signalling in this species is complex and involves a combination of behavioural traits such as lek attendance, fighting rate and lek centrality[30,31], alongside multiple sexual ornaments including blue chroma colouration[32], lyre size[30] and eye comb size[31]. These traits convey different aspects of male quality and integrate information over various timescales. Consequently, sexual signalling in the black grouse is multidimensional and dynamic, offering an exceptional opportunity to quantify the effects of deleterious mutations on multiple sexual traits in order to identify honest signals of genetic quality.

We combined whole genome sequencing data from 190 male black grouse with comprehensive individual-based data to investigate the genetic architecture of lifetime reproductive success. Our dataset comprises complete life histories for 168 'core males' captured as yearlings and incomplete histories for 22 'non-core males' captured as adults. Individual measures of annual mating success along with data on multiple behavioural and ornamental traits were gathered over a decade (2002–2012 inclusive) from five lekking sites in central Finland (Supplementary Fig. 1a). We aimed to (1) quantify the fitness effects of predicted deleterious mutations, including both homozygous and heterozygous mutations; (2) evaluate the effects of deleterious mutations across different genomic regions; and (3) isolate the direct and indirect pathways through which deleterious mutations influence male reproductive success, focusing on their effects on the expression of behavioural and ornamental traits.

## Results and discussion

Sequencing to an average coverage of 32× generated 2.41 billion 150-base pair (bp) paired-end Illumina sequencing reads, which were used to call 7,271,836 high-quality biallelic single nucleotide polymorphisms (SNPs). The study population showed little in the way of population structure and 97.5% of all pairs of individuals were unrelated (Supplementary Results and Discussion). Given the small proportion of related pairs of individuals in our dataset, we do not anticipate that relatedness structure will influence our results and conclusions.

### Inbreeding

We found clear evidence of inbreeding, with $F_{ROH}$ (the proportion of an individual's autosomal genome in runs of homozygosity (ROHs)) being non-zero across all individuals in the population (Fig. 1a). This is in line with previous observations suggesting that black grouse do not actively avoid inbreeding, although passive mechanisms such as female-biased dispersal[33] and the limited temporal overlap of related individuals due to sex-specific differences in lifespan[34] may reduce its occurrence. Inbreeding levels varied substantially among individuals, with $F_{ROH}$ ranging from 0.220 to 0.329 (Fig. 1a). The mean ROH length was 65 kilobases (kb), corresponding to an average of 346 autozygous SNPs, while the maximum ROH length was 29 megabases (Mb), corresponding to 189,221 autozygous SNPs. To investigate the antiquity of inbreeding, we classified ROHs into three length categories: short (<1 Mb), intermediate (1–2 Mb) and long (>2 Mb), which correspond to inbreeding events approximately >50, 25–50 and <25 generations ago, respectively (Fig. 1b). The vast majority of ROHs were short (*n* = 692,103) with relatively few intermediate (*n* = 1,781) and long (*n* = 505) ROHs being detected. Consequently, short ROHs contributed the most to $F_{ROH}$, indicating that inbreeding is mainly historical, dating back more than 50 generations or roughly 150 years ago, assuming a generation time of 3 years (ref. 35). However, long ROHs contributed disproportionately to $F_{ROH}$ in some of the most inbred individuals, occasionally spanning nearly entire scaffolds (Fig. 1c). This observed variation in inbreeding among individuals is a prerequisite for detecting inbreeding depression[36].

### Predicting deleterious mutations

We identified putatively deleterious mutations using two widely adopted approaches, evolutionary constraint and functional effect prediction, to evaluate whether they produce consistent insights. Evolutionary constraint was estimated using GERP++ (ref. 37), which quantifies the reduction in the number of substitutions at each nucleotide position throughout the genome compared to neutral expectations. Genomic evolutionary rate profiling (GERP) scores were assigned to a total of 6,954,487 SNPs residing on the 29 largest autosomal scaffolds (Methods) and ranged from −8.57 to 4.29, with higher GERP scores indicating greater evolutionary constraint. The distribution of GERP scores (Fig. 2a) was skewed towards lower values, with 52.4% of SNPs having scores below zero, which is a threshold commonly used to indicate neutral evolution[11,38]. To identify those mutations with predicted deleterious effects, we focused on the 413,489 SNPs (5.9%) assigned to the highest GERP score category (≥4), as these mutations are most likely to be deleterious, although they may not necessarily have the largest effect sizes[11].

To annotate SNPs according to their predicted effects on protein structure and function, we used SnpEff[12] to assign 6,375,440 autosomal SNPs to one of four non-mutually exclusive impact classes

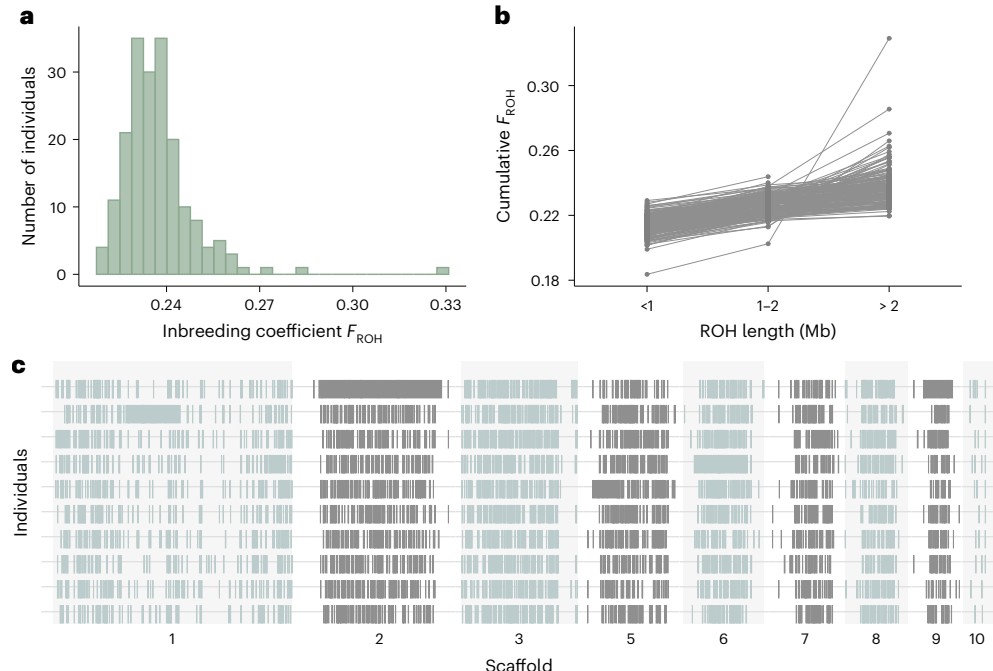

**Fig. 1 | Genomic inbreeding in male black grouse. a**, Histogram of genomic inbreeding ($F_{ROH}$) values across 190 individual males. **b**, The cumulative contribution to $F_{ROH}$ of ROHs shorter than the length indicated on the $x$ axis. Each line indicates a single individual ($n = 190$) and the value of $F_{ROH}$ on the right corresponds to the overall genomic inbreeding coefficient. **c**, The distribution of ROHs across the nine largest autosomal scaffolds. Each row represents a different individual, with the five most inbred individuals (that is, with the highest $F_{ROH}$ values) being shown above and the five least inbred individuals (that is, with the lowest $F_{ROH}$ values) being shown below. Individual scaffolds are indicated by alternating background shades. Scaffold 4 is not shown as this is sex-linked.

(low, moderate, high and modifiers; Fig. 2b). To identify those mutations with the strongest predicted deleterious effects, we focused on the 5,341 SNPs (0.08%) classified as 'high impact', which are assumed to have a high (disruptive) impact on the protein, including predicted lost start and stop codons, loss of function (LOF) mutations, gained stop codons and nonsense mediated decay mutations (Fig. 2c). Only 274 (5%) of these SNPs had a GERP score ≥4 and there was no evidence for a decline in average GERP scores with decreasing SnpEff impact category (Extended Data Fig. 1), echoing a previous study[39] that also found little overlap between mutations predicted to have large effect sizes through evolutionary conservation and functional prediction. The majority of mutations with GERP scores ≥4 and high-impact SnpEff mutations occurred at low frequencies in the population (Fig. 2d,e). The number of deleterious mutations identified by GERP and SnpEff is influenced by conceptual and methodological factors as described in the Supplementary Results and Discussion.

Next, we estimated individual genomic mutation loads by calculating the total number of derived deleterious mutations in each individual's genome while correcting for variation in genotyping success as described in the Methods. We further decomposed the total genomic mutation load of each individual into the 'homozygous load', comprising deleterious mutations in homozygosity, and the 'heterozygous load', comprising deleterious mutations in heterozygosity. This was implemented separately for mutations with GERP scores ≥4 (hereafter referred to as the 'GERP load') and mutations classified as being of high impact by SnpEff (hereafter referred to as the 'SnpEff load'). Individuals carried on average 120,796 (± 4,846 s.d., range 96,528–125,561) and 1,640 (± 105 s.d., range 1,235–1,793) mutations identified by GERP and SnpEff, respectively, with the number of mutations in heterozygosity being larger than the number of mutations in homozygosity (Fig. 2f,g). The total GERP and SnpEff loads were not significantly correlated (Pearson's $r = 0.13$, $P = 0.08$), suggesting that individuals with more mutations in evolutionarily conserved regions do not necessarily carry more mutations with large predicted functional effects.

The total, homozygous and heterozygous loads were approximately normally distributed (Extended Data Fig. 2a,b). As expected, $F_{ROH}$ was significantly positively associated with the homozygous load (GERP $r = 0.78$, $P < 0.001$; SnpEff $r = 0.28$, $P < 0.001$; Extended Data Fig. 2c) and significantly negatively associated with the heterozygous load (GERP $r = -0.77$, $P < 0.001$; SnpEff $r = -0.36$, $P < 0.001$; Extended Data Fig. 2c). However, no clear relationship was observed between $F_{ROH}$ and the total load (GERP $r = -0.01$, $P = 0.88$; SnpEff $r = -0.03$, $P = 0.63$; Extended Data Fig. 2c) indicating that, while $F_{ROH}$ can be used as proxy for an individual's homozygous load, it is not necessarily informative about an individual's total load.

**Fitness effects of genomic mutation loads and inbreeding**
To address a key knowledge gap concerning the fitness effects of predicted deleterious mutations, we constructed separate Bayesian generalized linear mixed effect models (GLMMs) of lifetime mating success (LMS), fitting either the total GERP load or the total SnpEff load as predictor variables together with core versus non-core male as a fixed effect and lekking site as a random effect (Methods). The posterior standardized $\beta$ estimates were negative for both the total GERP load (median $\beta$ estimate is −0.21, 95% credible interval (CI) = −0.27, −0.14; Fig. 3a, Extended Data Fig. 3a and Supplementary Tables 1 and 2) and the total SnpEff load (median $\beta$ estimate is −0.11, 95% CI = −0.18, −0.04; Fig. 3a, Extended Data Fig. 3b and Supplementary Tables 1 and 2). However, the negative association between the total SnpEff load and LMS was only present when mutations flagged with warning messages regarding the SnpEff database were excluded (with warning messages included, median $\beta$ estimate is −0.07, 95% CI = −0.15, 0.01). This suggests that the accuracy of the predictions of SnpEff depends on the quality of the reference genome and its annotation. The stronger negative effect of the total GERP load on LMS compared to the SnpEff load may be a reflection of the distinct properties of those mutations identified by each prediction approach and/or the number of deleterious mutations identified, as described in the Supplementary Results and Discussion.

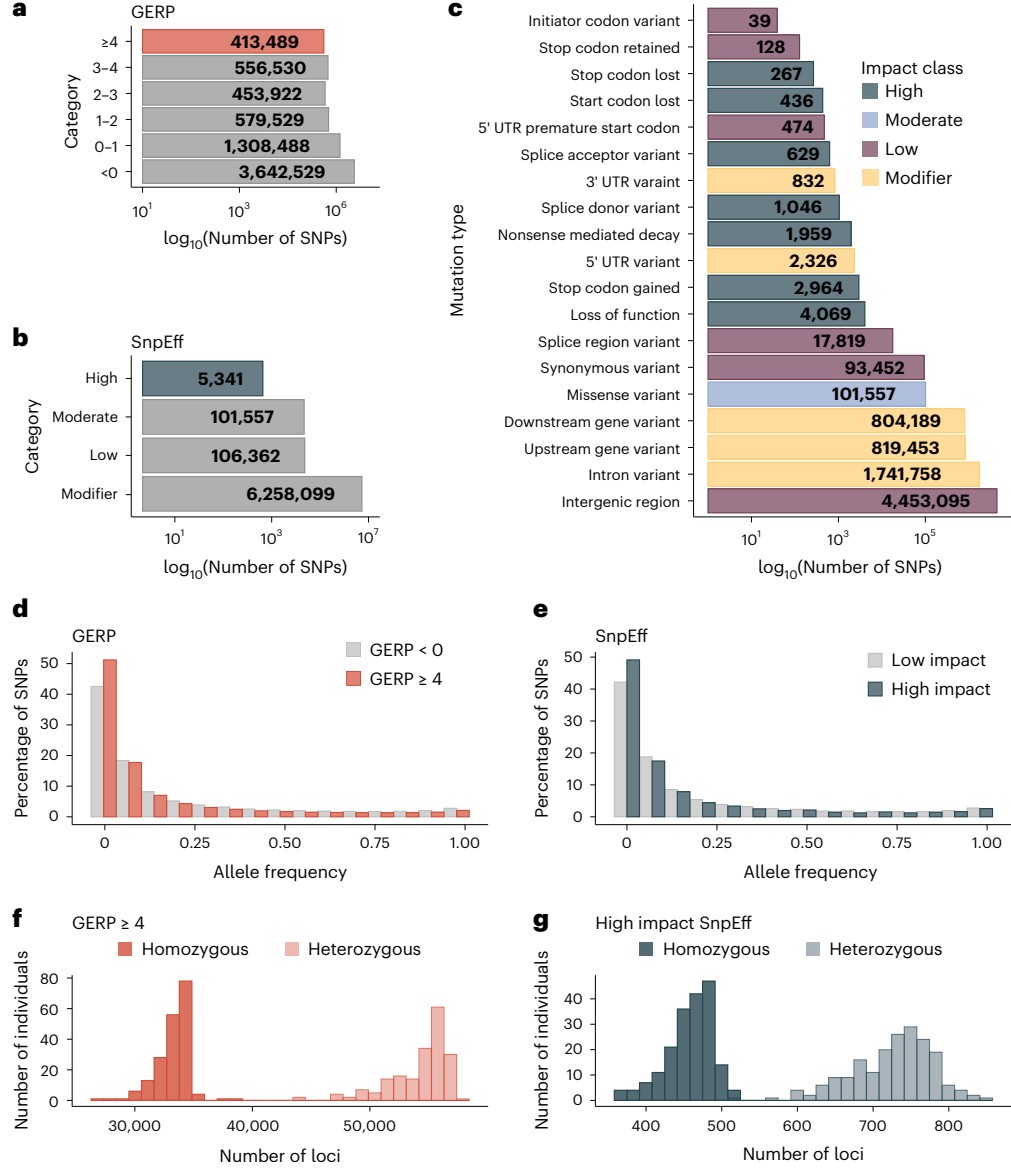

**Fig. 2 | Predicted deleterious mutations in male black grouse. a**, Bar plot showing the number of mutations assigned to each GERP score category. Mutations with the strongest predicted deleterious effects (that is, SNPs with GERP scores ≥4) are highlighted in red. **b**, Bar plot showing the number of mutations assigned to each SnpEff impact category. Mutations with the strongest predicted deleterious effects (that is, SNPs classified as 'high impact') are highlighted in dark blue. **c**, A detailed breakdown of the mutations annotated by SnpEff. UTR, untranslated region. **d**, Histogram of allele frequencies of derived putatively neutral (GERP scores <0) and highly deleterious (GERP scores ≥4) mutations. **e**, Histogram of allele frequencies of derived mutations classified by SnpEff as low and high impact. **f**, Histogram of the number of homozygous and heterozygous mutations with GERP scores ≥ 4 across all 190 individuals. **g**, Histogram of the number of homozygous and heterozygous mutations classified by SnpEff as high impact across all 190 individuals.

Our genomic mutation load estimates capture information on both homozygous and heterozygous mutations, both of which are expected to contribute to an individual's mutation load[20]. On the basis of this, we hypothesized that the total GERP and SnpEff loads would explain more variation in LMS than inbreeding. To test this, we constructed a GLMM of LMS with $F_{ROH}$ as a predictor variable together with the same fixed and random effects as described above. We found clear evidence of inbreeding depression as the posterior standardized $\beta$ estimates of $F_{ROH}$ were predominantly negative and their 95% CI did not overlap zero (median $\beta$ estimate is −0.14, 95% CI = −0.20, −0.07; Fig. 3a, Extended Data Fig. 3c and Supplementary Tables 1 and 2). In support of our hypothesis, the total GERP load accounted for more than twice the variation in LMS compared to $F_{ROH}$ (median marginal $r^2$ = 2.0% versus 0.8%, respectively; Supplementary Table 1). However, there was little difference in the explained variance

of the total SnpEff load (median marginal $r^2$ = 1.0; Supplementary Table 1) and $F_{ROH}$.

### Effects of the homozygous and heterozygous loads

To quantify the contributions of homozygous and heterozygous mutations to fitness, we constructed a GLMM of LMS in which the homozygous and heterozygous loads were fitted jointly as predictors together with the same fixed and random effects described above, separately for GERP and SnpEff. Including both load components together in a single model allowed us to quantify the fitness effects of each component while controlling for the other (Extended Data Fig. 4). We found that, regardless of the prediction approach, both the homozygous and heterozygous loads were negatively associated with LMS (Fig. 3b, Extended Data Fig. 3d,e and Supplementary Table 1). An effect of the homozygous load on fitness is to be expected given that deleterious

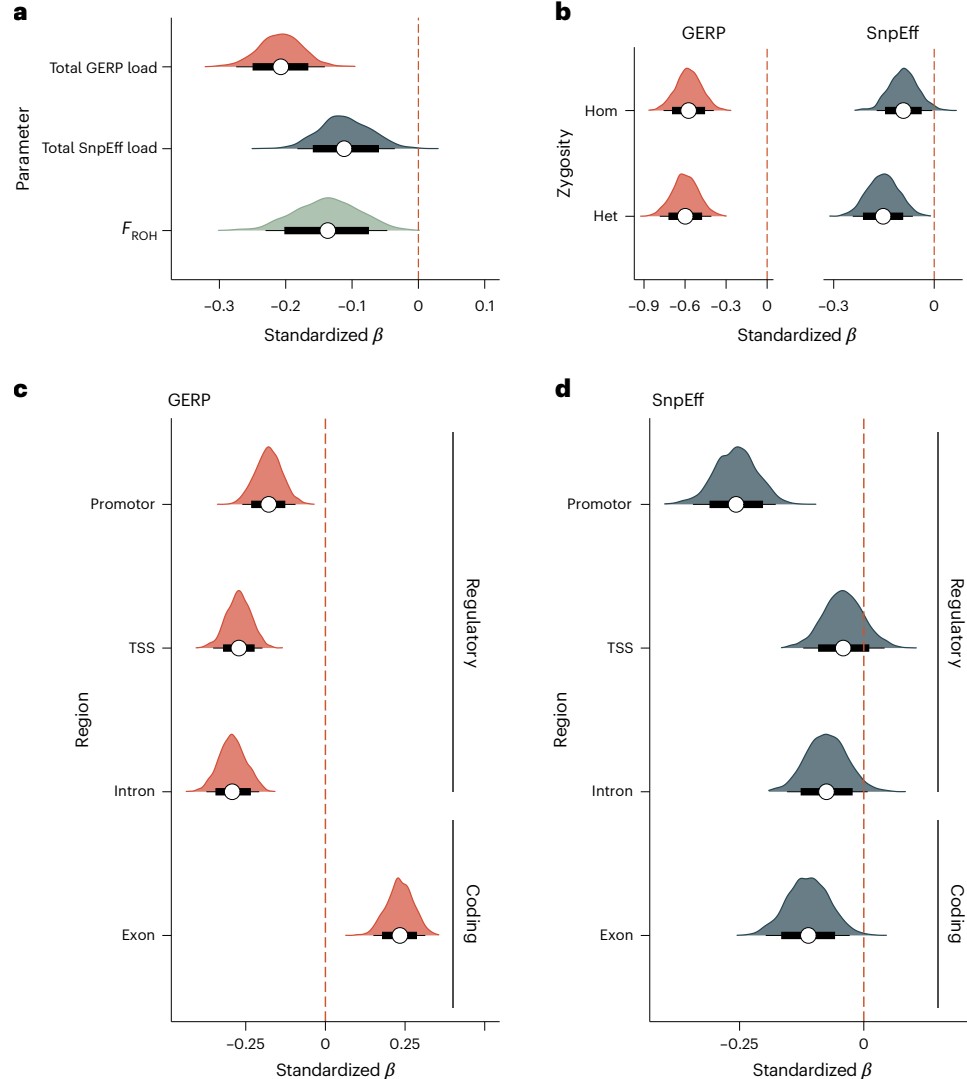

**Fig. 3 | Fitness effects of inbreeding and genomic mutation loads in male black grouse. a**, Posterior distributions of the standardized $\beta$ estimates of the total GERP load, the total SnpEff load and genomic inbreeding ($F_{ROH}$) on LMS. **b**, Posterior distributions of the standardized $\beta$ estimates of the homozygous and heterozygous GERP and SnpEff loads on LMS. **c**, Posterior distributions of

the standardized $\beta$ estimates of the total GERP load partitioned into mutations residing within regulatory and coding regions. **d**, Posterior distributions of the $\beta$ estimates of the total SnpEff load partitioned into mutations residing within regulatory and coding regions. The white circles represent the median posterior estimates, the thick black lines the 80% CIs and the thin black lines the 95% CIs.

mutations in homozygosity will be expressed regardless of their dominance coefficients. By contrast, an effect of the heterozygous load should only be found if the mutations in question are not completely recessive, which is the expectation for deleterious mutations with small to moderate effect sizes[40–42]. Additionally, we observed that the effect sizes of the homozygous and heterozygous GERP loads (median $\beta$ estimates are −0.57 and −0.60, respectively; Fig. 3b) were substantially more negative than the effect size of the total GERP load (median $\beta$ estimate is −0.21; Fig. 3a). This pattern probably arises because the total GERP load does not account for the strong opposing correlations of the homozygous and heterozygous GERP load with genomic inbreeding (Extended Data Figs. 2c and 4).

**Regulatory versus coding effects**

Both functional non-coding and protein-coding regions can be subject to purifying selection[43,44], although the former include various regulatory elements such as promoters, enhancers and silencers, which may experience different selective pressures, depending on their roles in gene regulation. To investigate whether the fitness effects of deleterious mutations differ by genomic region, we classified each

mutation according to its location within a promoter (excluding the transcription start site (TSS), $n = 16{,}493$ for GERP; $n = 1{,}151$ for SnpEff), TSS ($n = 2{,}408$ for GERP; $n = 913$ for SnpEff), intron ($n = 104{,}045$ for GERP; $n = 2{,}204$ for SnpEff) or exon ($n = 21{,}581$ for GERP; $n = 3{,}813$ for SnpEff). We then computed the total load separately for each genomic region and prediction approach, and used the resulting values as predictor variables in separate Bayesian GLMMs of LMS, while including the same fixed and random effects as described for the models above.

For both prediction approaches, the total load in promoter regions was negatively associated with LMS (Fig. 3c,d and Supplementary Tables 3 and 4). Furthermore, when controlling for the number of mutations, the $\beta$ estimates of the total SnpEff load in promoter regions were substantially more negative than the $\beta$ estimates of mutations in other regions (Supplementary Results and Discussion). Promoters, which facilitate transcription factor binding and initiate transcription, are crucial in regulating gene expression[45]. Additionally, mutations in highly conserved regulatory regions, which are often found near the promoters of genes involved in critical developmental processes[46–48], can have deleterious effects as conserved regulatory regions tend to stabilize gene expression more effectively than less conserved ones[49].

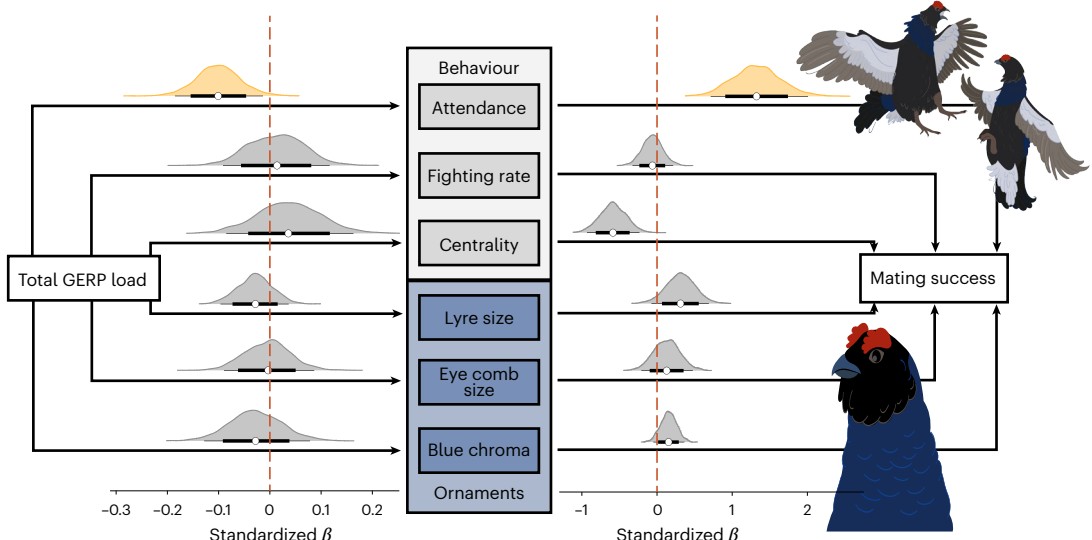

**Fig. 4 | The effects of deleterious mutations on sexual trait expression and mating success in male black grouse.** On the left are shown the posterior distributions of the standardized $\beta$ estimates of the total GERP load on three sexually selected behavioural traits (top) and three sexually selected ornamental traits (bottom). On the right are shown the posterior distributions of the $\beta$ estimates of the six sexual traits on AMS. The white circles represent the median posterior estimates, the thick black lines the 80% CIs and the thin black lines the 95% CIs. The significant indirect pathway of the total GERP load on AMS mediated by lek attendance is highlighted in yellow.

Finally, mutations in *cis*-regulatory regions are often codominant[25], suggesting that heterozygous mutations can have negative effects on fitness. This is consistent with the observed negative effects of the heterozygous GERP and SnpEff loads on LMS in the black grouse.

Results for TSS and intronic regions were more nuanced. The posterior $\beta$ estimates of the total load were mostly negative for both GERP and SnpEff, indicating a general trend towards deleterious effects (Fig. 3c,d and Supplementary Tables 3 and 4). However, the 95% CIs for SnpEff overlapped zero, indicating weaker, less reliable or less consistent negative associations for mutations predicted as high impact by SnpEff in these regions. When controlling for the number of mutations, the $\beta$ estimates of the total GERP load in the TSS were substantially more negative than those of mutations in other regions (Supplementary Results and Discussion). Mutations in the TSS are expected to be deleterious because they can impede RNA polymerase binding, reduce transcription initiation and decrease translation efficiency and messenger RNA stability[50]. Indeed, expression quantitative trait loci in model species are frequently located in or around the TSS[51–54], indicating that mutations in these regions can strongly impact gene expression and phenotypic variation. Intronic mutations, on the other hand, can be deleterious as they may disrupt gene splicing[55], which has been shown to have detrimental effects on disease traits[56].

The effects of exonic mutations on LMS varied, depending on the prediction approach (Fig. 3c,d and Supplementary Tables 3 and 4). While exonic mutations predicted by SnpEff were negatively associated with LMS, in line with theoretical expectations, exonic mutations with high GERP scores were positively associated with LMS. A potential explanation for this finding is that highly deleterious exonic GERP mutations may be eliminated by viability selection during early life stages, reducing embryonic or chick survival. Consequently, exonic GERP mutations surviving to adulthood may be less harmful or might even be beneficial owing to ongoing adaptation[10] or functional turnover[11], explaining their overall net-positive association with LMS.

The phenotypic effects of deleterious mutations might further depend on which genes they affect, and, consequently, which biological processes are disrupted. For instance, deleterious mutations in genes influencing male sexual traits, such as those related to immunity[28,57,58] androgen production[59] and oxidative stress[60,61], might be particularly relevant for male reproductive success. To investigate this, we used gene ontology annotations to identify subsets of deleterious mutations within genes associated with six biological processes hypothesized to be important for sexual signalling and sexual selection (Supplementary Table 5). We found that mutations in genes linked to specific processes including androgen metabolism, immunity and response to oxidative stress had negative effects on LMS (Supplementary Tables 6 and 7 and Extended Data Figs. 5 and 6), as described in the Supplementary Results and Discussion.

## Behavioural versus ornamental pathways

Little is known about how deleterious mutations impact fitness through their effects on various phenotypic traits at the organismal level. Sexually selected traits, because of their condition-dependence, may represent a large mutational target, as proposed by the 'genic capture hypothesis'[62,63], making them susceptible to the accumulation of genome-wide mutations. However, it remains unclear whether the mutation load affects male reproductive success directly or indirectly via its effects on sexual trait expression. To address this question, we used extensive, individual-based repeated measures of behavioural and ornamental traits collected on an annual basis to disentangle the direct and indirect effects of deleterious mutations on male reproductive success.

First, we tested for a direct effect of the total GERP load on annual mating success (AMS) by fitting it as a predictor variable in a Bayesian GLMM alongside six behavioural and ornamental traits: (1) lek attendance; (2) fighting rate; (3) lek centrality; (4) lyre size; (5) eye comb size; and (6) blue chroma. Fitting all of these predictors together in a single model allowed us to estimate the direct effect of the total GERP load on AMS while controlling for the mediating effects of the behavioural and ornamental covariates (Methods). As this model incorporates repeated individual measures from males attending different leks, we included a two-level fixed effect of age category (yearling versus adult) together with sampling year and ID nested within lek as random effects. Correcting for sampling year and lek further accounts for population fluctuations that could influence male–male competition, mate availability and mating success. We found no evidence of a direct effect of the total GERP load on AMS (median $\beta$ estimate is −0.13, 95% CI = −0.36, 0.11). Similarly, using the same model structure,

we found no significant direct effect of the total SnpEff load on AMS (median $\beta$ estimate is −0.11, 95% CI = −0.38, 0.16).

To investigate the indirect pathway(s) by which deleterious mutations affect AMS, we constructed separate Bayesian linear mixed effect models for each of the six behavioural and ornamental traits, fitting the total load as a predictor variable. Age category was again included as a fixed effect together with sampling year and ID nested within lek as random effects. We then quantified the indirect pathways using mediation analysis, where indirect effects were calculated as the product of the effect of the predictor (the total load) on the mediator (sexual trait) and the effect of the mediator on the response variable (AMS; Methods). We identified a single indirect pathway between the total GERP load and AMS mediated by lek attendance (median $\beta$ estimate is −0.13, 95% CI = −0.28, −0.01; Fig. 4 and Supplementary Tables 8, 9 and 11). No evidence was found of any indirect pathways linking the total SnpEff load to AMS (Extended Data Fig. 7 and Supplementary Tables 8, 10 and 11).

For male black grouse, high lek attendance is critical for achieving reproductive success, as those males with the highest attendance typically secure the most central territories and engage more frequently in energetically costly fights[30,64]. Furthermore, a males' current lekking performance is correlated with his past lekking effort[65,66]. Consequently, high lek attendance not only indicates short-term body condition and display effort, but also reflects longer term, cumulative reproductive effort[65,66], which are traits expected to be influenced by genome-wide deleterious mutations. The indirect pathway between the total GERP load and AMS mediated by lek attendance therefore supports previous studies of black grouse[30,64] and other lekking species[67], highlighting the critical role of lek attendance as a determinant of male mating success.

## Conclusions

Computational predictions of deleterious mutations are increasingly used to evaluate extinction risks in vulnerable species[68] and to optimize conservation strategies[69]. However, our understanding of the fitness effects of these mutations and how they influence key life-history traits related to survival and reproduction remains limited. We addressed this knowledge gap by integrating genomic and life-history data from the black grouse, an iconic lekking species. Four important results were obtained. First, two independent prediction approaches, evolutionary conservation and functional prediction, both identified deleterious mutations associated with reduced male lifetime reproductive success. This congruence of results effectively validates both approaches, although lack of reliance of GERP on functional annotations may offer advantages in non-model organisms, where gene annotations are often incomplete or suboptimal.

A second key insight was that mutations contributing to individual fitness in male black grouse are not limited to homozygous deleterious mutations; heterozygous deleterious mutations also negatively impact mating success, as pointed out by Morton et al.[20] almost 70 years ago. By implication, considering only homozygous mutations[39,70,71], risks underestimating the true realized load due to the exclusion of partially recessive mutations expressed in the heterozygous state. The relative fitness contributions of homozygous and heterozygous mutations are expected to vary across species depending on factors such as genetic architecture[72], dominance relationships[73] and species-specific evolutionary histories[74]. Hence, further research is needed to evaluate the effects of homozygous and heterozygous mutations across a broad range of taxa and ecological contexts[75].

Third, our results indicate that deleterious mutations located in promoters (including the TSS) have disproportionally negative effects on male reproductive success. This observation is consistent with findings from model systems, where regulatory mutations have been shown to have major impacts on ecologically relevant traits[25,76,77], disrupt the stabilization of gene expression[49] and reduce translation

efficiency[50]. This suggests that regulatory elements should be given more attention in studies of wild populations. Furthermore, it opens up an exciting research opportunity to explore how deleterious mutations in promotors affect gene regulatory networks and the ability of individuals to regulate gene expression to maximize their fitness.

Finally, our study uncovered a specific biological pathway through which deleterious mutations appear to affect male mating success in the black grouse. Specifically, the total GERP load reduces lek attendance, a crucial behavioural trait for mating success and an indicator of genetic quality in several lekking species[78,79]. This suggests that deleterious mutations in evolutionary conserved regions mainly influence reproductive outcomes in this species through behavioural changes rather than by altering the expression of sexual ornaments. Indeed, behavioural traits in black grouse are under constant sexual selection throughout life, in contrast to ornamental traits[80,81], which are strongly age-dependent and experience selection mainly in older males[26]. As lekking behaviour captures variation in both short- and long-term reproductive investment, which is highly dependent on body condition[30,65], our results are consistent with the genic capture hypothesis, which posits that sexually selected traits reflect genetic variation in condition influenced by genome-wide mutations[63]. By contrast, ornamental traits might be governed by specific genes, such as those impacting the efficiency of metabolic pathways that convert dietary carotenoids to red pigments[82,83], as well as by epigenetic mechanisms that are sensitive to age[84], genetic quality[84] and environmental factors[85].

In conclusion, sexual signalling depends upon the intricate coordination of multiple traits that are simultaneously expressed, requiring mechanisms finely tuned to an individual's resource availability and allocation. Gene regulatory mechanisms enable dynamic gene expression, allowing organisms to adapt their phenotypes to context-dependent needs, which vary throughout their lifespan. Disruptions to these mechanisms are therefore likely to be detrimental to fitness, as our findings demonstrate. This crucial insight into the genetic architecture of male reproductive success not only advances our understanding of sexual selection but may also enhance genomics-guided conservation efforts by highlighting the pivotal role of regulatory regions in determining individual fitness.

## Methods

All fieldwork was ethically approved by the Central Finland Environmental Centre (permissions KSU-2003-L-25/254 and KSU-2002-L4/254).

### Data and sample collection

Life-history data and blood samples were collected from 190 male black grouse between 2002 and 2012 inclusive from five study sites in Central Finland (Supplementary Fig. 1a). For 171 males that were first caught as yearlings[86], hereafter referred to as 'core males' (data partially published in ref. 86), complete life histories were obtained as previously described[26] while for the remaining 26 individuals (part of ref. 87), life histories were incomplete as these animals were not captured as yearlings. Morphological measures were taken before the lekking season (January–March) by capturing the birds in walk-in traps baited with oats[64,80]. The individuals were aged as yearlings or older on the basis of their plumage characteristics[88]. Lyre length was measured to the nearest 1.0 mm as the length of the longest outer tail feather from the base to the tip. Pictures of both eye combs were taken with a scale held behind the bird's head as a standard. The combined area of both eye combs were measured (in cm²) using ImageJ[89] and the sum of these measurements was used for analysis. Next, a representative breast feather was collected to quantify blue chroma reflection using a Avantes Spectrophotometer (GS 3100, EG & G Gamma Scientific) as described by ref. 32. All the individuals were marked with aluminium tarsus rings carrying unique serial numbers as well as with three colour rings to facilitate identification from a

distance. Blood (1–2 ml) was sampled from the brachial vein using a heparinized syringe. Red blood cells were stored after centrifugation in 70% ethanol at 4 °C.

During the main lekking season (end of March to April) of each year, the birds were observed from before sunrise (02:00–04:00) until they departed the lek (08:00–09:00) using binoculars and telescopes from hides located close to the leks. Male mating success was quantified as the number of observed copulations with females on leks. LMS was defined as the total number of observed copulations throughout the lifespan of each individual male. LMS is a strong predictor of male fitness as females generally mate once with a single male[90], observed copulations are highly concordant with true parentage inferred from genetic data[90], and infertile clutches are rare[91]. Furthermore, although male fitness is determined by additional factors such as clutch size, hatching success and chick survival, sexual selection on male genetic quality is likely to be strongest during precopulatory stages, as hatching success and chick survival are predominantly dependent on environmental factors[92].

Males were assumed to have died when they were never caught or sighted subsequently. Male lekking behaviour was recorded using scan sampling every 5th to 20th minute. The behaviours that were recorded included (1) attendance (that is, presence on the lek); (2) fighting rate; and (3) lek centrality. Lek attendance was calculated in proportion to the highest attending male on the lek in that year. Fighting rate was calculated as the percentage of scans when a male was observed performing this behaviour. Lek centrality was measured as the distance to the lek centre, calculated using a $10 \times 10$ m$^2$ grid system on each lek. Each males' position was mapped to the closest 1 m on the grid and the median of all mapped points was taken as his distance to the lek centre. The centre of individual male territories was determined as the median of all coordinates recorded per male during a given mating season, and the overall lek centre was determined as the median of all of the coordinates recorded during that mating season. Thus, lower lek centrality values are indicative of more centrally displaying males.

### DNA extraction and sequencing

Genomic DNA was extracted from red blood cells using either a Qiagen Blood and Tissue Extraction Kit (162 individuals) or a standard chloroform-isoamylalcohol protocol (28 individuals). Library preparation was performed at the Beijing Genomics Institute as described in the Supplementary Methods. The 150-bp paired-end sequencing reads were generated on a DNBSEQ-G400 platform. The adaptor sequences were subsequently removed and low-quality reads and contamination were excluded using SOAPnuke[93]. Low-quality reads were defined as reads with more than 40% of bases with a quality value below ten. If a read contained any Ns, the entire read was discarded. The quality of the raw sequence data was checked using FastQC v.0.11.9 (ref. 94).

### Genotyping

Before genotyping, we scaffolded and annotated an existing black grouse reference genome assembled by the 10K Bird Project (B10K)[95,96] as described in the Supplementary Methods. The quality filtered reads were then aligned to the genome using the Burrows–Wheeler alignment (BWA-mem) algorithm v.0.7.13 (ref. 97). The resulting SAM files were converted into binary format and subsequently sorted and indexed using samtools v.1.15.1 (ref. 98). SNPs were genotyped using the mpileup algorithm from BCFtools v.1.11 (ref. 98), requiring a minimum quality of 20 (-q 20) and the mapping quality of reads with excessive mismatches was downgraded (-C 50). The range of the mean coverage across individuals per partially filtered SNP was 0.005–368×, whereas the range of the mean coverage across SNPs per individual was 22–33×. SNPs were further filtered using VCFtools v.0.1.17 (ref. 99) for a minimum depth of 20× (--minDP 20), a maximum of 30% missing data (--max-missing 0.7), a maximum mean depth of twice the mean depth (--max-meanDP 60) and a minimum quality score of 30 (--minQC 30).

Additionally, only biallelic SNPs were retained (--min-alleles 2, --max-alleles 2) and indels were discarded (--remove-indels).

### Population structure and relatedness

To characterize the study population, we tested for population genetic structure using PLINK v.1.90 (ref. 100) and quantified genetic differentiation by calculating $F_{ST}$ values among all pairs of leks using VCFtools v.0.1.17 (ref. 99). We also quantified patterns of pairwise genomic relatedness among individuals using NgsRelate v.2 (ref. 101) and PLINK[100] as explained in Supplementary Methods.

### Runs of homozygosity

ROHs were inferred using the --roh algorithm implemented in BCFtools[102]. This algorithm detects regions of autozygosity using a hidden Markov model that assesses the likelihood of the two alleles at a given locus being identical by descent. The accuracy of $F_{ROH}$ estimation with BCFtools therefore does not depend on the settings of sliding window parameters[102] used in other commonly used ROH detection software like PLINK[103]. Before ROH calling, we did not filter the dataset for Hardy–Weinberg equilibrium (HWE), minor allele frequency (MAF) or linkage disequilibrium (LD), as this has been shown to have little impact on ROH calling performance but substantially reduces dataset size[103]. We only used genotypes with a minimum quality of 30 (--G30) to identify autozygous regions with the default allele frequency settings. The BCFtools output was filtered for ROHs that were at least 100-kb long and contained a minimum of 100 SNPs. We then calculated each individual's genomic inbreeding coefficient, $F_{ROH}$, as the proportion of the autosomal genome in ROHs[104]. ROHs were divided into three length categories: short (<1 Mb), intermediate (1–2 Mb) and long (>2 Mb). ROH lengths were converted to generations ago using the following equation[105]:

$$L = \frac{100}{2 \times g}$$

where $L$ represents the ROH length measured in centimorgans and $g$ represents the number of generations ago. To convert ROH length in base pairs to centimorgans, we assumed the ratio of genetic to physical distance to be 1 cM:1 Mb (ref. 106). To convert generations into calendar years, we assumed a generation time of 3 years for the black grouse[35]. Our code for visualizing ROHs was adapted from ref. 107.

### Predicting deleterious mutations

We estimated evolutionary conservation across the genome using GERP++ (ref. 37). This software takes a multispecies alignment file as input, evaluates the reduction in the number of substitutions compared to neutral expectations, and subsequently calculates a GERP score for each position, with higher GERP scores indicating greater evolutionary conservation. To generate a multispecies alignment, we used the publicly available multi-alignment file of 363 avian genomes (https://cgl.gi.ucsc.edu/data/cactus/363-avian-2020.hal, downloaded on 16 October 2023) in HAL format[108] published by the Bird 10K consortium[96] as a starting point, after which we used the Progressive Cactus toolkit v.2.6.12 (ref. 109) to edit the HAL file to our specific requirements.

First, we reduced the multiple alignment file to a total of 72 genomes using the halRemoveSubtree and halRemoveGenome commands, excluding species in the Neoaves clade from the phylogenetic tree. Next, we added the black grouse and the white-tailed ptarmigan (*Lagopus leucura*, NCBI RefSeq assembly GCF_019238085.1) reference genomes to the multiple alignment using the add branch command, resulting in a phylogenetic tree consisting of 74 genomes (Supplementary Fig. 2) with a total branch length of 5.19 substitutions per site. The resulting HAL file was converted to MAF format per scaffold using the command cactus-hal2maf. We estimated the branch-lengths of the updated phylogenetic tree with iqtree v.2.2.6 (ref. 110) using a

concatenation of 5,000 random 1-kb windows, while using a topology created by TimeTree as a constraint. The windows were restricted to non-coding regions with a minimum of 70 aligned genomes and were extracted using a combination of functions from Progressive Cactus[109], maffilter[111] and SeqKit[112]. GERP++ was subsequently used to calculate expected and observed substitution rates per scaffold. We excluded the Z chromosome from our analysis, which comprises 7.5% of the total genome length. We also excluded the black grouse genome from the GERP score calculation by using the -j flag within the gerpcol command. GERP scores were calculated on the basis of the 29 largest autosomal scaffolds only because (1) these scaffolds comprise the majority (97.4%) of the total autosomal genome length; (2) this measure increased computational efficiency both at the HAL to MAF file conversion step and for the calculation of the GERP scores because both commands are executed per scaffold; and (3) among-species coverage is expected to be lower for smaller scaffolds, potentially resulting in lower GERP scores. A custom bash script was used to subset the GERP scores calculated throughout the entire genome to include only locations corresponding to the filtered SNP dataset described above, using the BEDOPS toolkit v.2.4.41 (ref. [113]) and the intersect command from bedtools v.2.27.1 package[114]. We did not filter the SNPs for HWE, MAF or LD as this could lead to the exclusion of rare, highly deleterious variants, resulting in genomic mutation loads being underestimated.

### SnpEff
We predicted the effects of genetic variants using SnpEff v.5.2 (ref. [12]) with a custom SnpEff database built for the black grouse. Coding regions and genes were extracted from the black grouse gene annotation in GFF format using the gff3_to_fasta function from the GFF3 toolkit (https://github.com/NAL-i5K/GFF3toolkit). Protein sequences were inferred using the agat_sp_extract_sequences function in AGAT[115]. We then built the custom database using the build command of SnpEff. SnpSift[12] was subsequently used to filter the database for high-impact SnpEff mutations, defined as those classes of mutation that are assumed to have disruptive effects on the protein − for example, due to protein truncation, loss of function or because the mutation triggers nonsense mediated decay[12]. We excluded SnpEff annotations from further analyses if they contained any kind of warning message, for example regarding the genome annotation. For comparability with the GERP results, we focused on SNPs residing only on the largest 29 autosomal scaffolds.

### Genome polarization
We polarized the black grouse genome using the reconstructed genome of the most recent common ancestor of the black grouse and the white-tailed ptarmigan, which was generated by Progressive Cactus (see above). The white-tailed ptarmigan is a small, non-lekking grouse species that diverged from the black grouse around 7.1 million years ago[116]. Nucleotide differences between the black grouse and the common ancestor were exported from the HAL alignment described above using the halSnps command[109] and the ancestral allele was subsequently appended to the SnpEff-annotated VCF file using the vcf-annotate command from VCFtools v.0.1.16 (ref. [99]). Lastly, where the reference allele in the VCF differed from the inferred ancestral allele, we adjusted the genotypes of both alleles accordingly using the jvarkit java-based utility set v.1.1.0 (ref. [117]) so that the ancestral allele was encoded as 0 and the derived allele was encoded as 1.

### Estimating individual genomic mutation loads
We estimated each individual's total, homozygous and heterozygous load based on the mutations identified by GERP and SnpEff, respectively, focussing on derived mutations with large predicted disruptive effects. For the former, we focused on mutations with GERP scores ≥4, which are collectively referred to as the 'GERP load'. For the latter, we focused on mutations identified by SnpEff as being of 'high impact',

which are collectively referred to as the 'SnpEff load'. For both prediction approaches, we calculated the total, homozygous and heterozygous load of each individual as follows:

$$\text{Total load}_{[ij]} = \frac{L_{\text{HM}} + 0.5 L_{\text{HT}}}{L_{\text{T}}}$$

$$\text{Homozygous load}_{[ij]} = \frac{L_{\text{HM}}}{L_{\text{T}}}$$

$$\text{Heterozygous load}_{[ij]} = \frac{L_{\text{HT}}}{L_{\text{T}}}$$

where $L_{\text{HM}}$ is the total number of homozygous derived loci in category $j$ in individual $i$; $L_{\text{HT}}$ is the number of heterozygous derived loci in category $j$ in individual $i$; and $L_{\text{T}}$ is the total number of loci genotyped in category $j$ in individual $i$.

### Modelling the effects of predicted deleterious mutations on fitness
We tested for differences in genomic inbreeding and individual genomic mutation loads among leks by constructing linear models of $F_{\text{ROH}}$, the total GERP load and the total SnpEff load with lekking site included as a fixed effect predictor variable. We found no significant differences in the total load between any pairs of leks, but a significant difference in $F_{\text{ROH}}$ between one pair of sites (Supplementary Table 12). Therefore, we included lek as a random effect in all our statistical models to control for differences in genomic inbreeding as well as potential lek-specific environmental or demographic differences that might influence the modelled traits.

Next, to evaluate the effects of individual genomic mutation loads on LMS, we constructed Bayesian GLMMs using the R package brms v.2.19.0 (ref. [24]). Beforehand, we tested for zero inflation in a frequentist null model of LMS with the testZeroInflation function in DHARMa[118]. As the result was statistically significant ($P < 2.2 \times 10^{-16}$), we used a zero-inflated Poisson distribution for all models of LMS. Models were constructed separately for the total GERP load and the total SnpEff load. Following previous studies (for example, refs. [119–121]), these models assumed the additivity of deleterious mutations, where the ancestral allele is expected to only partially suppress the expression of the derived allele in the heterozygous state. Thus, both homozygous and heterozygous mutations were considered to contribute towards the total mutation load. In these models, the z-transformed total load was included as a fixed effect, with mutations in the homozygous state contributing twice as much as mutations in the heterozygous state, reflecting the number of alleles that contribute towards the total mutation load. We also included core versus non-core male as a two-level fixed effect and lek as a random effect in these models. Afterwards, we repeated the models while fitting the z-transformed homozygous load and the z-transformed heterozygous load together as predictors for both prediction approaches. Finally, as the number of deleterious mutations identified by GERP and SnpEff differed substantially, we compared their effect sizes on LMS while controlling for the number of mutations as described in the Supplementary Methods.

### Testing for the effects of mutations in different genomic regions
Next, we annotated each mutation to determine whether it overlapped a TSS, promoter, intron and/or exon using R packages GenomicFeatures v.1.42.3 (ref. [122]) and rtracklayer v.1.50.0 (ref. [123]). We defined a promoter as the region located between 2,000-bp upstream and 200-bp downstream of the annotated starting position of the genes[124]. A TSS was defined as being located between 300-bp upstream and 50-bp downstream of the gene's starting position[125]. If a mutation was found within a TSS, it was also inherently located within the promoter

region; therefore, we annotated it solely as being located in the TSS to avoid redundancy. Next, we calculated the total GERP load and the total SnpEff load separately for each of the four genomic regions and constructed eight Bayesian GLMMs of LMS as described above, one for each genomic region and prediction approach. The total mutation loads were again *z*-transformed and the same controlling variables and random effect structure were used as described above. Finally, as the number of deleterious mutations varied among different genomic regions, we compared their effect sizes on LMS while controlling for the number of mutations as described in the Supplementary Methods.

### Mediation analysis

To investigate whether deleterious mutations affect male reproductive success directly or indirectly via the expression of behavioural and/or ornamental traits, we performed a mediation analysis in two consecutive steps. First, we constructed six separate Bayesian LMMs testing for the effects of the total load on lek attendance, fighting rate, lek centrality, lyre size, eye comb size and blue chroma. We included age as a two-level fixed effect (yearling versus adult) and sampling year and ID nested within lekking site as random effects in these models. In the second step, we constructed a single Bayesian GLMM of AMS that included the six sexual traits as well as the total load as fixed effects. All seven variables were *z*-transformed to allow the computation of their relative contributions towards AMS. We again included age category as a fixed effect and sampling year plus ID nested within lekking site as random effects, while using a zero-inflated Poisson distribution as described above. Fitting all the traits in a single model allowed us to isolate the effect of each trait on reproductive success while controlling for the effects of the other traits. We then calculated the direct and indirect effects of the total load on AMS using the product method[126]. Specifically, we estimated indirect effects as the product of the effect of the predictor (that is, the total load) on the mediator (that is, the sexual trait) and the effect of the mediator on the response variable (that is, AMS). The direct effect was estimated as the effect of the predictor on the response variable, adjusted for the effects of the mediators. This analysis was implemented separately for the total GERP load and the total SnpEff load.

All the Bayesian models described in Methods were run for one million iterations using four independent Markov chains, with a thinning interval of 1,000 and a burn-in period of 500,000 iterations. We used generic weakly informative priors for the population-level effects (normal distribution mean = 0, s.d. = 1) and tested for prior sensitivity by repeating all models with the default brms priors and with an alternative prior specification (population-level effects mean = 10, s.d. = 10; intercept mean = 30, s.d. = 10) to ensure that our conclusions were not biased by the specified priors. Model performance was diagnosed by analysing divergent transitions, convergence, autocorrelation, *R* hat statistics and effective sampling sizes using the R package bayesplot v.1.10.0 (ref. 127). For each model, Bayesian versions of $R^2$ were calculated using the r2_bayes function from the performance package v.0.12.3 (ref. 128). A result was considered to be statistically significant if the 95% CI of the $\beta$ estimate did not overlap zero[129]. The full model outputs of the Bayesian GLMMs, including estimates for all of the fixed and random effects, can be found in the github repository (https://github.com/rshuhuachen/ms_load_grouse) under output/intervals.

All statistical analyses were implemented in R v.4.4.1 (ref. 130) using Rstudio v.2023.12.1.402 (ref. 131) and the results were visualized using the R packages ggplot2 v.3.4.4 (ref. 132), cowplot v.1.1.1 (ref. 133), bayesplot v.1.10.0 (ref. 127) and ggridges v.0.5.4 (ref. 134). The majority of bioinformatic workflows were integrated into Snakemake v.7.14 (ref. 135) using a conda environment with Anaconda v.23.7.4 (ref. 136) for enhanced reproducibility[137].

### Reporting summary

Further information on research design is available in the Nature Portfolio Reporting Summary linked to this article.

## Data availability

All code, phenotypic data and individual genomic mutation load estimates, as well as the genome annotation, are available via Zenodo at https://doi.org/10.5281/zenodo.15608151 (ref. 138). All sequencing data (SRA Study SRP499251 with BioAccession numbers SRR28526036–SRR28526225), the reference genome (GCA_043882375.1) and the RNA-seq data used for the genome annotation (SRA BioAccession no. SRR28789699) can be found under NCBI BioProject PRJNA1085187.

## Code availability

The code used for data analysis and creating the figures is available via Zenodo at https://doi.org/10.5281/zenodo.15608151 (ref. 138). The summarized code for the main analyses in RMarkdown-style can be found in html format at https://rshuhuachen.github.io/ms_load_grouse/.

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

## Acknowledgements

We thank the many field assistants and researchers who collected the samples and field data, with a special thanks to C. Lebigre, E. Virtanen, M. Kervinen, G. Ludwig, M. Halonen and J. Niva. We are grateful to R. V. Alatalo for establishing the black grouse project and to H. Siitari for continuing it after R. V. Alatalo's retirement and passing. We would further like to thank the B10K project for providing us with their unpublished, unscaffolded version of the black grouse reference genome. Lastly, we are grateful to K. Firsova, who produced the black grouse artwork used in our figures. This research was supported by a Deutsche Forschungsgemeinschaft grant (project no. 454606304 awarded to J.I.H.). The fieldwork was previously funded by the Academy of Finland (grant no. 7119165 awarded to H. Siitari) and the Finnish Centre of Excellence in Evolutionary Research (grant no. 7211271 to R. V. Alatalo).

## Author contributions

J.I.H., C.D.S. and K.v.O. conceived the study, acquired funding and supervised the research. C.D.S. provided the samples and life-history data. R.S.C. extracted and quality checked the DNA before sequencing. R.S.C. and K.H. designed the bioinformatic workflows for quantifying individual genomic mutation loads, while R.S.C. executed all the other data analyses with feedback from J.I.H. and K.H. The manuscript was drafted and revised by R.S.C. and J.I.H. All authors commented upon and approved the final manuscript.

## Funding

## Competing interests

The authors declare no competing interests.

## Additional information

**Extended data** is available for this paper at https://doi.org/10.1038/s41559-025-02802-8.

**Correspondence and requests for materials** should be addressed to R. S. Chen.

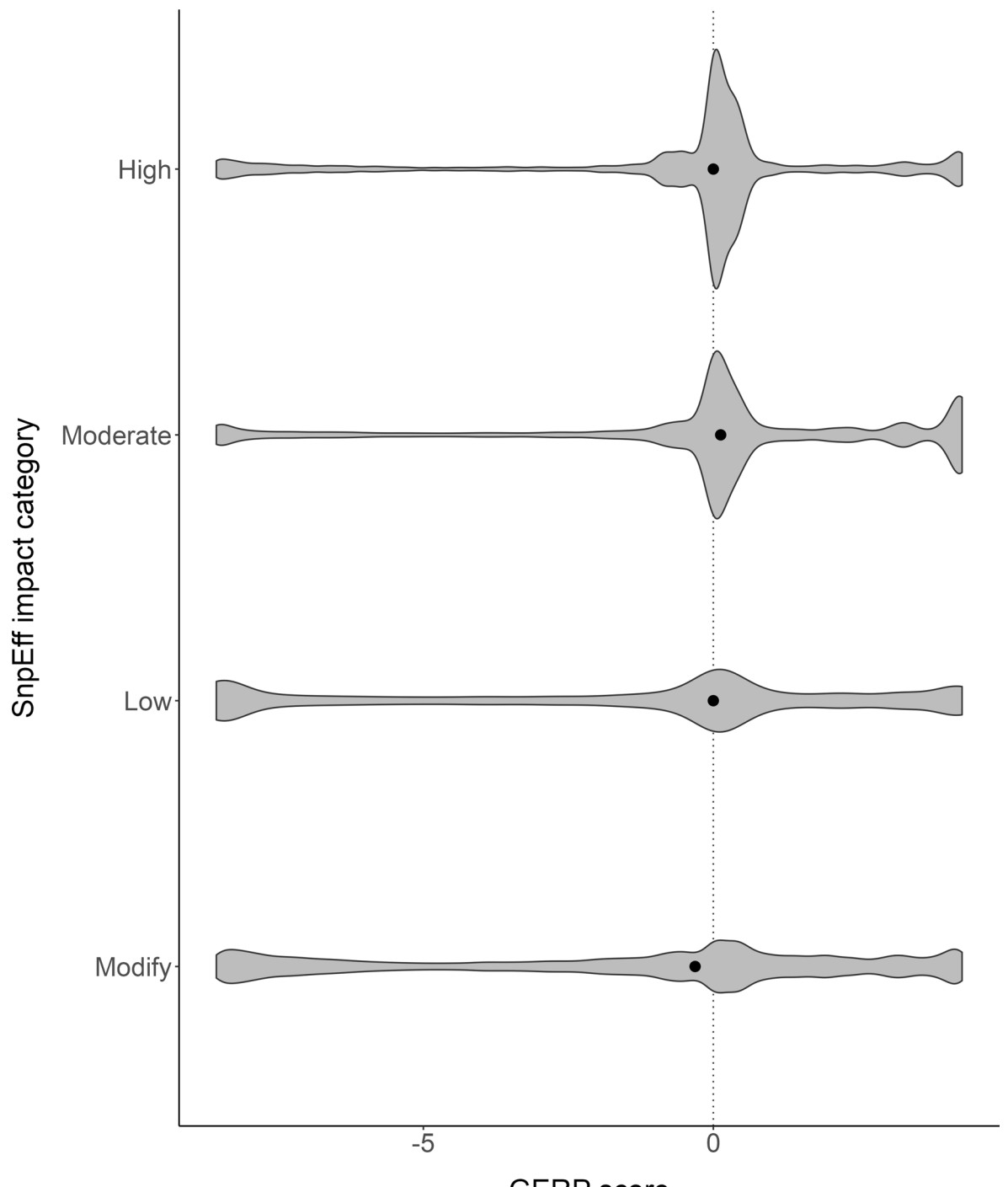

**Extended Data Fig. 1 | Violin plots showing the GERP score distributions of the mutations annotated by SnpEff, broken down by SnpEff impact category.** The black points indicate the median GERP scores of the mutations in each respective impact category.

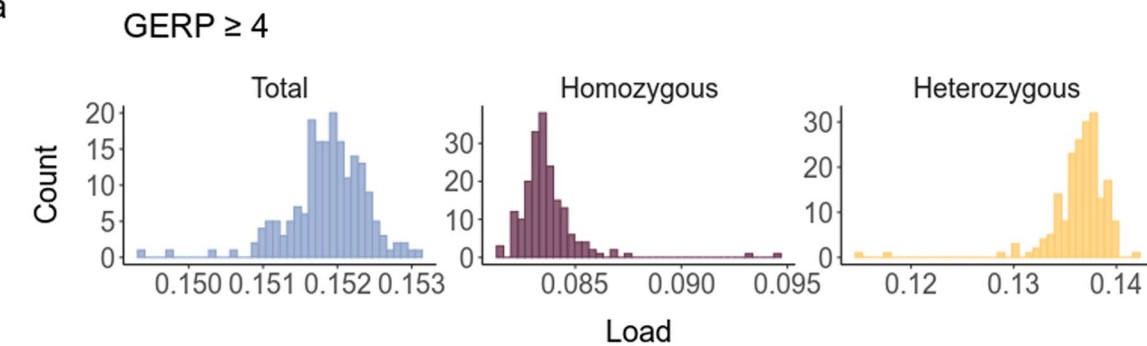

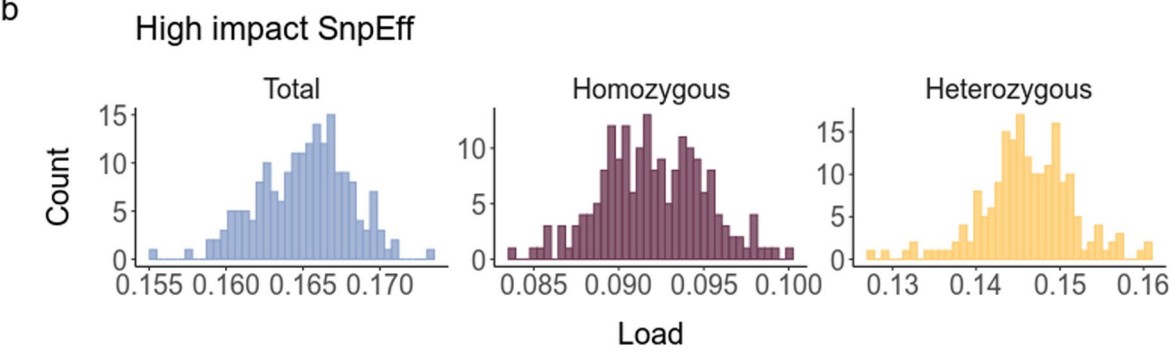

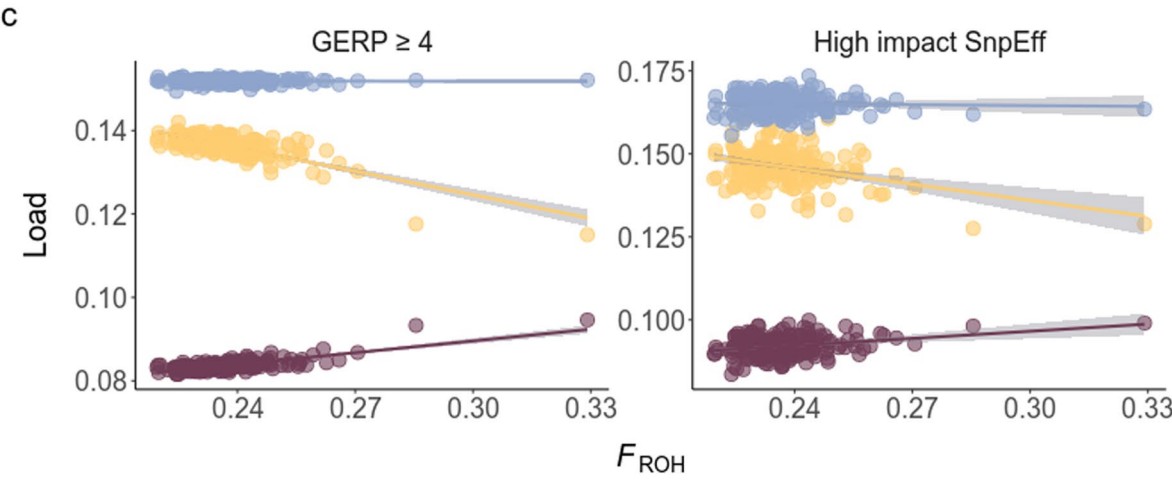

**Extended Data Fig. 2 | Distributions of individual genomic mutation loads and their relationships with genomic inbreeding.** Histograms of the total, homozygous and heterozygous load for (**a**) mutations with GERP scores ≥ 4 and (**b**) high impact SnpEff mutations. Panel (**c**) shows the relationships between $F_{ROH}$ and the three load components separately for GERP and SnpEff mutations. The lines indicate linear regressions between $F_{ROH}$ and the mutation load values, with the grey shaded areas representing the associated 95% confidence intervals.

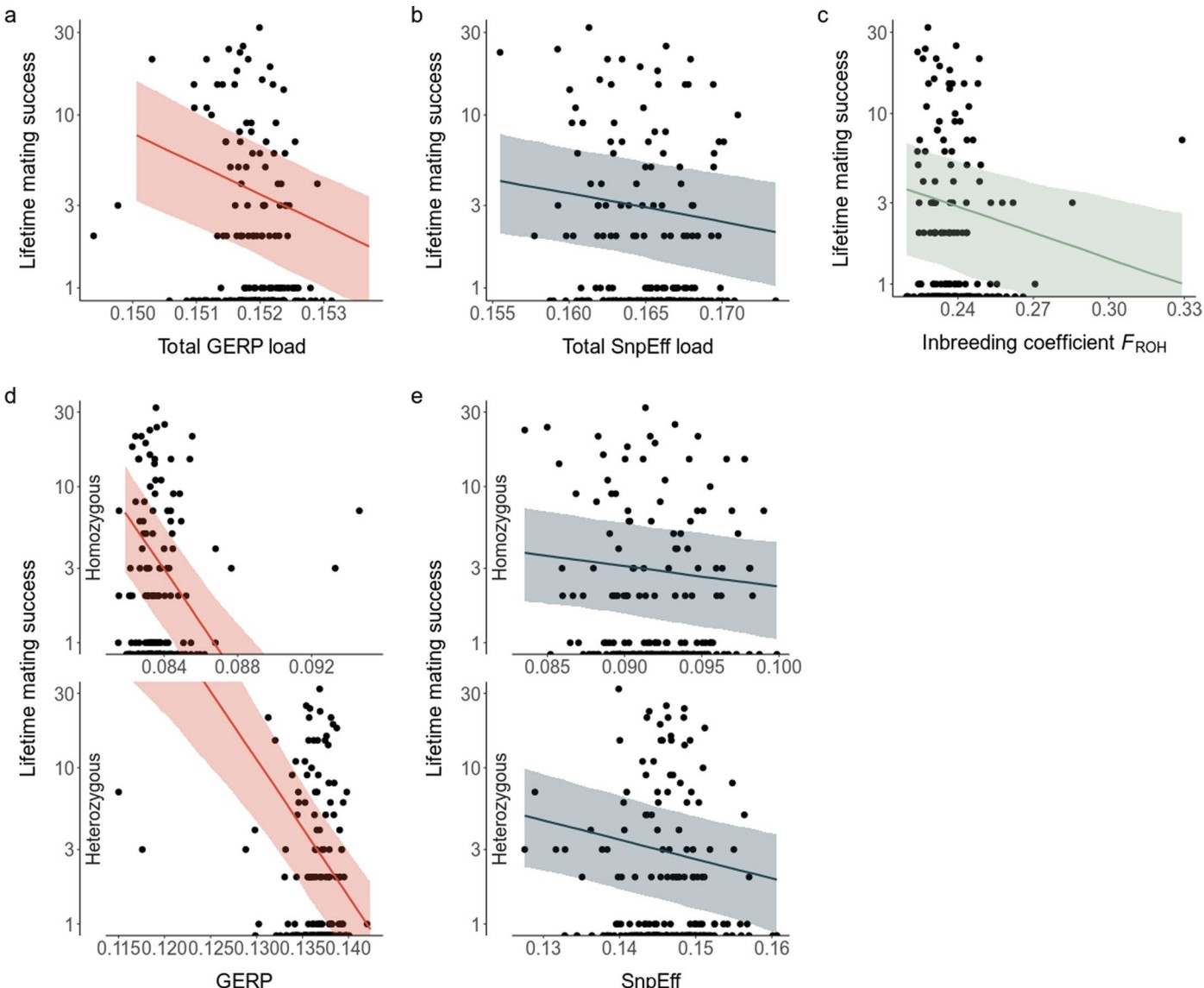

**Extended Data Fig. 3 | Raw data plots showing the fitness effects of inbreeding and genomic mutation loads in male black grouse.** Regression lines represent the predicted values of the response from Bayesian generalized linear mixed models (GLMMs) of lifetime mating success. The shaded areas represent the lower and upper bounds of the 95% uncertainty interval of the response. Results are shown separately for (**a**) the total GERP load; (**b**) the total SnpEff load; (**c**) $F_{ROH}$; (**d**) the homozygous and heterozygous GERP loads; and (**e**) the homozygous and heterozygous SnpEff loads.

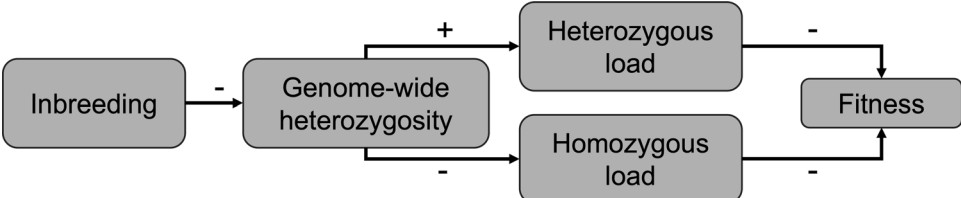

**Extended Data Fig. 4 | Directed acyclic graph illustrating theoretical relationships among inbreeding, genome-wide heterozygosity, the homozygous and heterozygous mutation loads, and fitness.** A minus (-) denotes a negative causal relationship and a plus (+) denotes a positive causal relationship. Increased inbreeding reduces genome-wide heterozygosity. Genome-wide heterozygosity is positively associated with the heterozygous load, but negatively associated with the homozygous load. Both load components are theoretically expected to reduce fitness. To isolate the individual effects of each load component on fitness, both the heterozygous and homozygous load must be included in the same statistical model to account for their association.

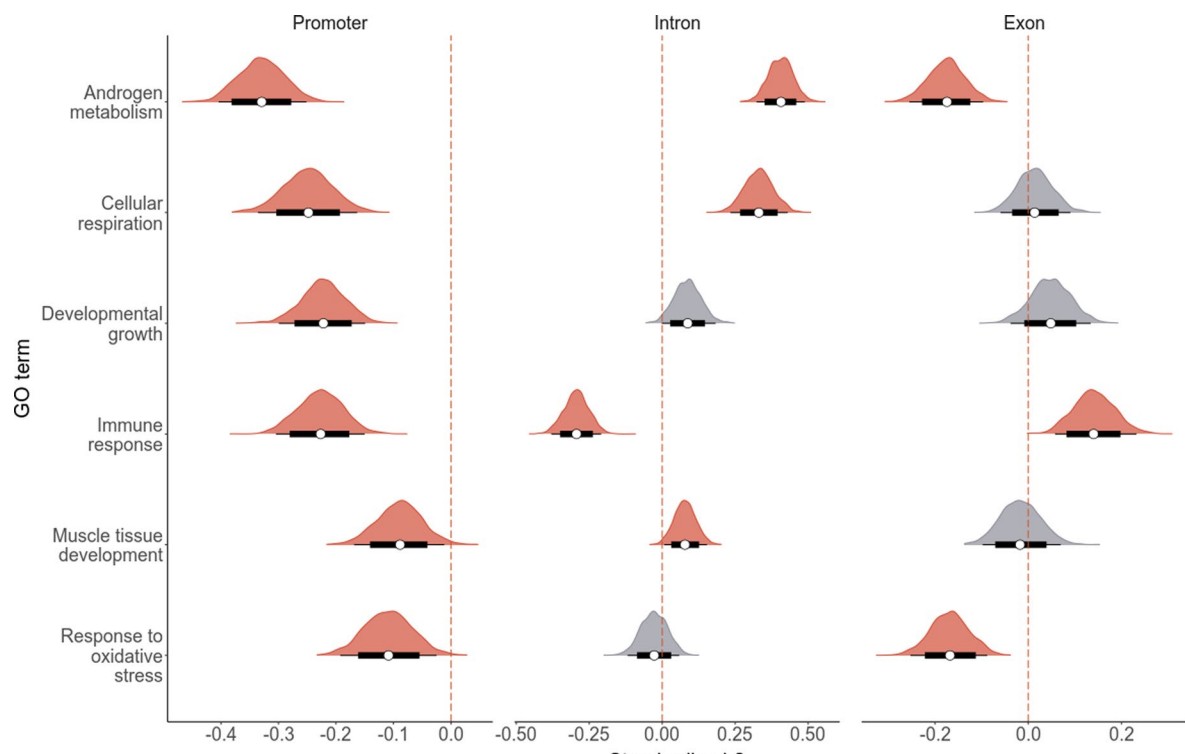

**Extended Data Fig. 5 | The effect of the total GERP load on lifetime mating success (LMS) for candidate GO terms, stratified by genomic region.** Bayesian generalized linear mixed models (GLMMs) of LMS were constructed separately for mutations located in the promoters, introns and exons (left to right) of genes associated with six GO terms (top to bottom) hypothesized to impact male reproductive success in the black grouse (see Supplementary Table 5 for details of our specific hypotheses and rationale). Shown are the posterior distributions of the $\beta$ estimates of the total GERP load on LMS. The white circles represent the median posterior estimates, the thick black lines the 80% CIs and the thin black lines the 95% CIs. Distributions highlighted in red are considered statistically significant (that is the 95% CIs do not overlap zero).

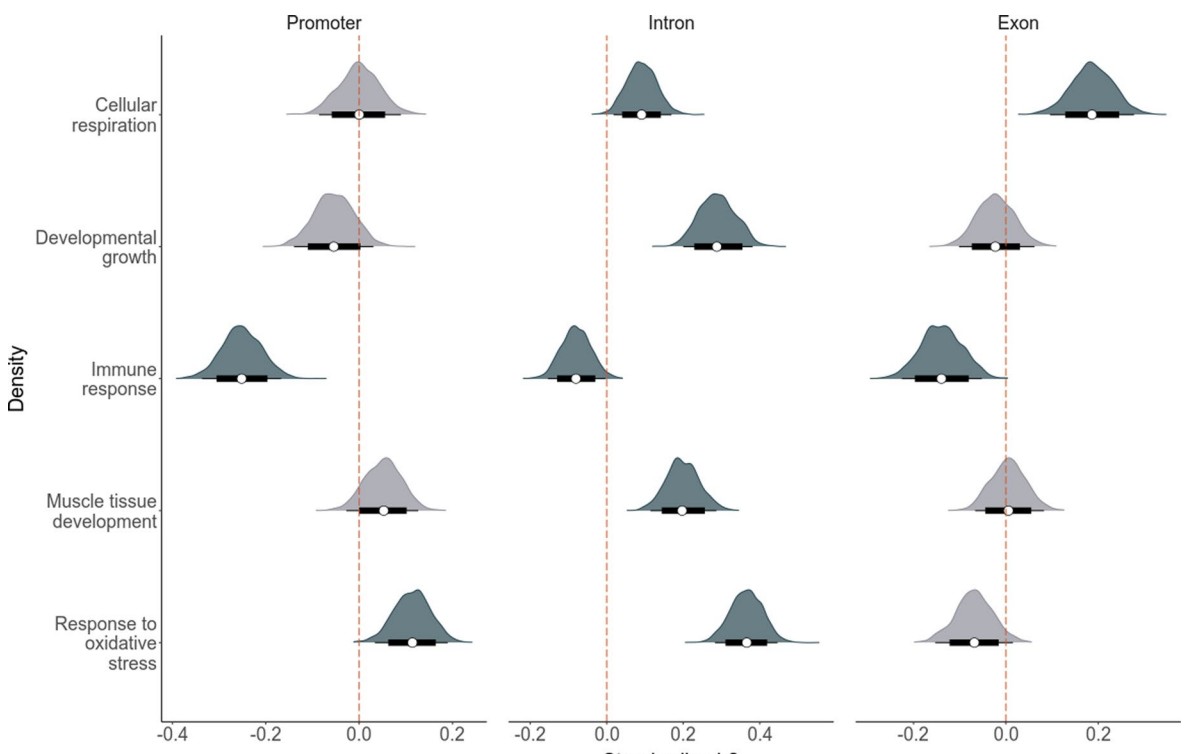

**Extended Data Fig. 6 | The effect of the total SnpEff load on lifetime mating success (LMS) for candidate GO terms stratified by genomic region.** Bayesian generalized linear mixed models (GLMMs) of LMS were constructed separately for mutations located in the promoters, introns and exons (left to right) of genes associated with five GO terms (top to bottom) hypothesized to impact male reproductive success in the black grouse (see Supplementary Table 5 for details of our specific hypotheses and rationale). Shown are the posterior distributions of the $\beta$ estimates of the total SnpEff load on LMS. The white circles represent the median posterior estimates, the thick black lines the 80% CIs and the thin black lines the 95% CIs. Distributions highlighted in dark grey are considered statistically significant (that is the 95% CIs do not overlap zero).

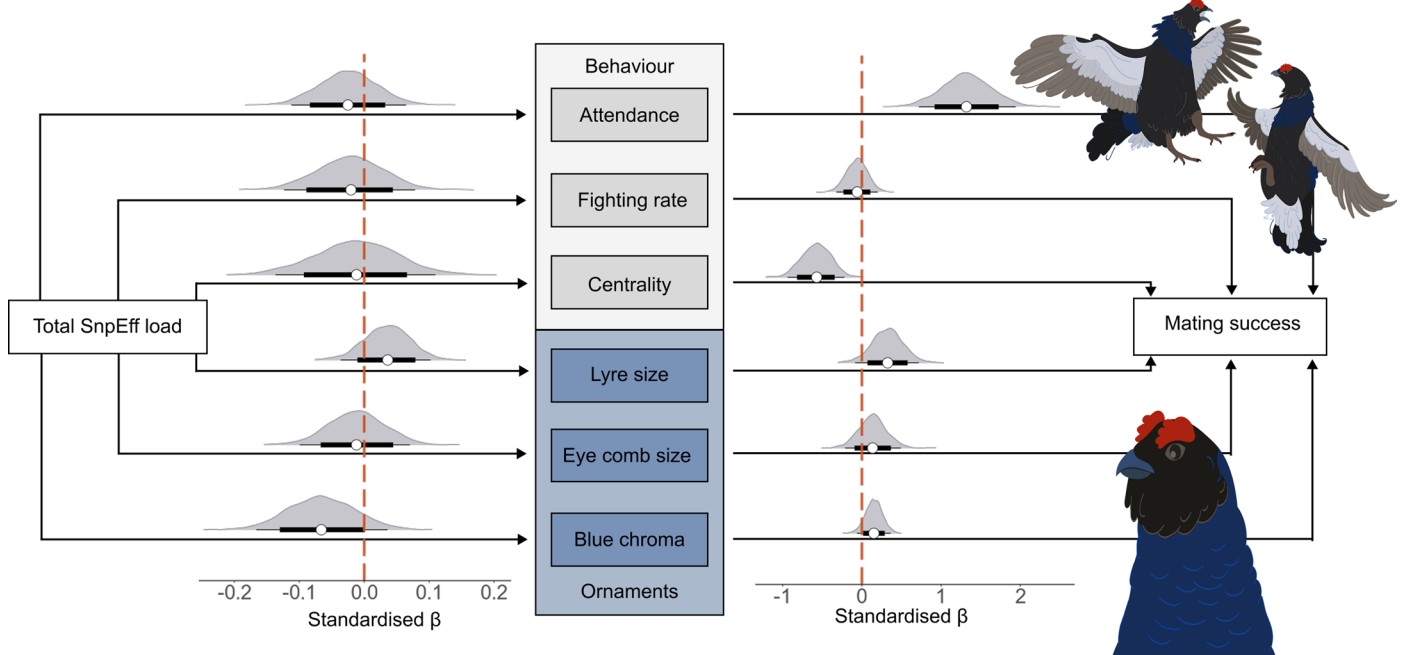

**Extended Data Fig. 7 | The effects of the total SnpEff load on sexual trait expression and annual mating success (AMS) in male black grouse.** On the left are shown the posterior distributions of the standardised $\beta$ estimates of the total SnpEff load on three sexually selected behavioural traits (top) and three sexually selected ornamental traits (bottom). On the right are shown the posterior distributions of the standardised $\beta$ estimates of the six sexual traits on AMS. The white circles represent the median posterior estimates, the thick black lines the 80% CIs and the thin black lines the 95% CIs.

# Reporting Summary

## Statistics

For all statistical analyses, confirm that the following items are present in the figure legend, table legend, main text, or Methods section.

| n/a | Confirmed | |
|---|---|---|
| ☐ | ☒ | The exact sample size ($n$) for each experimental group/condition, given as a discrete number and unit of measurement |
| ☐ | ☒ | A statement on whether measurements were taken from distinct samples or whether the same sample was measured repeatedly |
| ☐ | ☒ | The statistical test(s) used AND whether they are one- or two-sided<br>*Only common tests should be described solely by name; describe more complex techniques in the Methods section.* |
| ☐ | ☒ | A description of all covariates tested |
| ☐ | ☒ | A description of any assumptions or corrections, such as tests of normality and adjustment for multiple comparisons |
| ☐ | ☒ | A full description of the statistical parameters including central tendency (e.g. means) or other basic estimates (e.g. regression coefficient) AND variation (e.g. standard deviation) or associated estimates of uncertainty (e.g. confidence intervals) |
| ☐ | ☒ | For null hypothesis testing, the test statistic (e.g. $F$, $t$, $r$) with confidence intervals, effect sizes, degrees of freedom and $P$ value noted<br>*Give P values as exact values whenever suitable.* |
| ☐ | ☒ | For Bayesian analysis, information on the choice of priors and Markov chain Monte Carlo settings |
| ☒ | ☐ | For hierarchical and complex designs, identification of the appropriate level for tests and full reporting of outcomes |
| ☐ | ☒ | Estimates of effect sizes (e.g. Cohen's $d$, Pearson's $r$), indicating how they were calculated |

*Our web collection on statistics for biologists contains articles on many of the points above.*

## Software and code

Policy information about availability of computer code

| Data collection | All data are available via the provided links (see "Data availability") |
|---|---|
| Data analysis | All code are available via the provided links (see "Code availability") |

For manuscripts utilizing custom algorithms or software that are central to the research but not yet described in published literature, software must be made available to editors and reviewers. We strongly encourage code deposition in a community repository (e.g. GitHub). See the Nature Portfolio guidelines for submitting code & software for further information.

## Data

Policy information about availability of data

All manuscripts must include a data availability statement. This statement should provide the following information, where applicable:
- Accession codes, unique identifiers, or web links for publicly available datasets
- A description of any restrictions on data availability
- For clinical datasets or third party data, please ensure that the statement adheres to our policy

All code, phenotypic data and individual genomic mutation load estimates as well as the genome annotation are available via https://github.com/rshuhuachen/ms_load_grouse. All sequencing data (SRA Study SRP499251 with BioAccession Numbers SRR28526036 – SRR28526225), the reference genome (GCA_043882375.1), and the RNA-Seq data used for the genome annotation (SRA BioAccession Number SRR28789699) can be found under NCBI BioProject PRJNA1085187.

# Research involving human participants, their data, or biological material

Policy information about studies with human participants or human data. See also policy information about sex, gender (identity/presentation), and sexual orientation and race, ethnicity and racism.

| | |
|---|---|
| Reporting on sex and gender | *Use the terms sex (biological attribute) and gender (shaped by social and cultural circumstances) carefully in order to avoid confusing both terms. Indicate if findings apply to only one sex or gender; describe whether sex and gender were considered in study design; whether sex and/or gender was determined based on self-reporting or assigned and methods used.*<br>*Provide in the source data disaggregated sex and gender data, where this information has been collected, and if consent has been obtained for sharing of individual-level data; provide overall numbers in this Reporting Summary. Please state if this information has not been collected.*<br>*Report sex- and gender-based analyses where performed, justify reasons for lack of sex- and gender-based analysis.* |
| Reporting on race, ethnicity, or other socially relevant groupings | *Please specify the socially constructed or socially relevant categorization variable(s) used in your manuscript and explain why they were used. Please note that such variables should not be used as proxies for other socially constructed/relevant variables (for example, race or ethnicity should not be used as a proxy for socioeconomic status).*<br>*Provide clear definitions of the relevant terms used, how they were provided (by the participants/respondents, the researchers, or third parties), and the method(s) used to classify people into the different categories (e.g. self-report, census or administrative data, social media data, etc.)*<br>*Please provide details about how you controlled for confounding variables in your analyses.* |
| Population characteristics | *Describe the covariate-relevant population characteristics of the human research participants (e.g. age, genotypic information, past and current diagnosis and treatment categories). If you filled out the behavioural & social sciences study design questions and have nothing to add here, write "See above."* |
| Recruitment | *Describe how participants were recruited. Outline any potential self-selection bias or other biases that may be present and how these are likely to impact results.* |
| Ethics oversight | *Identify the organization(s) that approved the study protocol.* |

Note that full information on the approval of the study protocol must also be provided in the manuscript.

# Field-specific reporting

Please select the one below that is the best fit for your research. If you are not sure, read the appropriate sections before making your selection.

☐ Life sciences   ☐ Behavioural & social sciences   ☒ Ecological, evolutionary & environmental sciences

For a reference copy of the document with all sections, see nature.com/documents/nr-reporting-summary-flat.pdf

# Ecological, evolutionary & environmental sciences study design

All studies must disclose on these points even when the disclosure is negative.

| | |
|---|---|
| Study description | Mutation load effects on sexually selected traits in black grouse |
| Research sample | A population of black grouse males (190 individuals) sampled in Central Finland. For the purpose of our study, this population is representative of the species. |
| Sampling strategy | See the manuscript |
| Data collection | See the manuscript |
| Timing and spatial scale | See the manuscript |
| Data exclusions | No data were excluded |
| Reproducibility | All code and data are publicly available so the study can be reproduced. |
| Randomization | No experiments were conducted to randomize, but relevant covariates like sampling site and sampling year have been statistically controlled for as described in the manuscript. |
| Blinding | Not applicable (no blinding performed) |

Did the study involve field work?   ☒ Yes   ☐ No

# Field work, collection and transport

| | |
|---|---|
| Field conditions | Data were collected from male black grouse in Central Finland. Behavioural data were collected from hides. See manuscript for details. |
| Location | Central Finland, see Supplementary Materials for exact locations |
| Access & import/export | All data were collected under the relevant permits and Finnish laws as described in the manuscript |
| Disturbance | Disturbance to the birds was minimised by entering the hides before the birds arrived on the leks and leaving after birds have left. |

# Reporting for specific materials, systems and methods

We require information from authors about some types of materials, experimental systems and methods used in many studies. Here, indicate whether each material, system or method listed is relevant to your study. If you are not sure if a list item applies to your research, read the appropriate section before selecting a response.

## Materials & experimental systems

| n/a | Involved in the study |
|---|---|
| ☒ | ☐ Antibodies |
| ☒ | ☐ Eukaryotic cell lines |
| ☒ | ☐ Palaeontology and archaeology |
| ☐ | ☒ Animals and other organisms |
| ☒ | ☐ Clinical data |
| ☒ | ☐ Dual use research of concern |
| ☒ | ☐ Plants |

## Methods

| n/a | Involved in the study |
|---|---|
| ☒ | ☐ ChIP-seq |
| ☒ | ☐ Flow cytometry |
| ☒ | ☐ MRI-based neuroimaging |

## Animals and other research organisms

Policy information about studies involving animals; ARRIVE guidelines recommended for reporting animal research, and Sex and Gender in Research

| | |
|---|---|
| Laboratory animals | No |
| Wild animals | Yes, black grouse as described in the manuscript. Species: Lyrurus tetrix, ages 1-6. Animals were released at the same location they were caught. |
| Reporting on sex | All analyses were performed on black grouse males only as described in the manuscript |
| Field-collected samples | Birds were captured and handled at the study location without transporting the animals to the lab |
| Ethics oversight | Birds were captured under permissions of the Central Finland Environmental Centre (permissions KSU-2003-L-25/254 and KSU-2002-L4/254). No official authorization was required for the sacrifice of the black grouse embryo used for generating RNAseq data for genome annotation, as declared by the Bielefeld University Animal Welfare Officer. |

Note that full information on the approval of the study protocol must also be provided in the manuscript.

## Plants

| | |
|---|---|
| Seed stocks | *Report on the source of all seed stocks or other plant material used. If applicable, state the seed stock centre and catalogue number. If plant specimens were collected from the field, describe the collection location, date and sampling procedures.* |
| Novel plant genotypes | *Describe the methods by which all novel plant genotypes were produced. This includes those generated by transgenic approaches, gene editing, chemical/radiation-based mutagenesis and hybridization. For transgenic lines, describe the transformation method, the number of independent lines analyzed and the generation upon which experiments were performed. For gene-edited lines, describe the editor used, the endogenous sequence targeted for editing, the targeting guide RNA sequence (if applicable) and how the editor was applied.* |
| Authentication | *Describe any authentication procedures for each seed stock used or novel genotype generated. Describe any experiments used to assess the effect of a mutation and, where applicable, how potential secondary effects (e.g. second site T-DNA insertions, mosiacism, off-target gene editing) were examined.* |

