## [Peer Review File · Nature Ecology & Evolution]

Predicted deleterious mutations reveal the genetic architecture of male reproductive success in a lekking bird

Corresponding Author: Ms Rebecca Chen

Version 0:

Decision Letter:

24th January 2025

Dear Rebecca,

Your manuscript entitled "Predicted deleterious mutations reveal the genomic mechanisms underlying fitness variation in a lekking bird" has now been seen by three reviewers, whose comments are attached. The reviewers have raised a number of concerns which will need to be addressed before we can offer publication in Nature Ecology & Evolution. We will therefore need to see your responses to the criticisms raised and to some editorial concerns, along with a revised manuscript, before we can reach a final decision regarding publication.

We therefore invite you to revise your manuscript taking into account all reviewer and editor comments. Please highlight all changes in the manuscript text file in Microsoft Word format.

* If you have not done so already please begin to revise your manuscript so that it conforms to our Article format instructions at <http://www.nature.com/natecolevol/info/final-submission>. Refer also to any guidelines provided in this letter.

* Extended Data Figures - please ensure that any supplementary figures and tables that are crucial to the manuscript's conclusions are converted into Extended Data figures and tables to increase visibility of these data. Extended Data figures and tables are online-only (present in the online PDF and full-text HTML versions of the paper), peer-reviewed display items that provide essential background to the article but are not included in the main article due to space constraints. A maximum of ten Extended Data display items (figures and tables) is permitted.

Link Redacted

Nature Ecology & Evolution is committed to improving transparency in authorship. As part of our efforts in this direction, we are now requesting that all authors identified as 'corresponding author' on published papers create and link their Open Researcher and Contributor Identifier (ORCID) with their account on the Manuscript Tracking System (MTS), prior to acceptance. ORCID helps the scientific community achieve unambiguous attribution of all scholarly contributions. You can create and link your ORCID from the home page of the MTS by clicking on 'Modify my Springer Nature account'. For more information please visit www.springernature.com/orcid.

[redacted]

Reviewer expertise:

Reviewer #1: population genomics, including genetic load

Reviewer #2: genetic load and sexual selection

Reviewer #3: population genetics, including genetic load

Reviewers' comments:

Reviewer #1 (Remarks to the Author):

Chen et al. present a highly interesting manuscript and contribute with valuable empirical data to a field that yet remains to be explored. I really enjoyed reading this manuscript and found it well-written and sound, and I was really impressed by the quality of the data set. However, the manuscript feels somewhat overloaded with content that needs to be out in a more extensive theoretical background, so a few changes would hopefully make it even more interesting. These are my main comments:

- regarding (iii) elucidate the pathways by which deleterious mutations influence reproductive success, focusing on their effects on the expression of behavioural and ornamental traits., I feel like this part is lacking in theoretical context in the introduction, even though it is quite extensively discussed. This part would benefit from clearer theoretical predictions to make it feel less descriptive.
- the manuscript mentions the value of this study for conservation. However, the applied link needs to be clarified or removed
- the value of understanding these links is unquestionable, but the part about optimising conservation strategies and improving captive breeding needs to be more thoroughly explained, or consider removing it.
- on that note, what is known about the inbreeding level in the study population? There is indeed data to explore ROH and testing if the mutational load increases with inbreeding levels is highly relevant for this study.
- reference genome: if I understand correctly, all processing has been done with a black grouse reference genome. Where is this individual from? Can the selection of a con-specific reference genome introduce a bias in mutational classification?
- exploring the gene ontology could be important to understand the homozygous and heterozygous load and the link to LMS. Had that been done?
- L123: what range of coverage? Did you exclude any due to low coverage?
- L386 and onwards: the use of core males is motivated to explore the sexual selection perspective, however an important part of the story goes missing without the females. Do you have samples from them? What do you know about their mutational load?
- is the study population temporally stable or fluctuating?

Reviewer #2 (Remarks to the Author):

This study has a great dataset of life-long fitness estimates which is very valuable for understanding genetic load. Bioinformatic works are very well done. However, the results of statistical tests are not sufficient to reach the major goal of linking genetic load to life-time reproductive success and the contradiction in the main results is unneglectable. Please find major comments below:

1. The structure of Introduction can be improved so that the readers can follow more easily. Now the Introduction starts with the fitness effects of genetic load (or mutation load), then goes to technical prospect of measuring genetic load with genomic data, and then goes back to the genetic principle of genetic load. I'd suggest the Introduction to be organized following the order of theory (how genetic load works and linkage to sexual selection) – technic (genomics) – study case (black grouse, why it is suitable, and research questions).

2. I'd suggest to measure and present genomic heterozygosity. It is a basic measurement reflecting "genetic quality" of an individual. The connection between genetic diversity (often measured as heterozygosity), genetic load and fitness has been widely discussed in population genetic and conservation genetics. It would be interesting to have a deeper look with such good dataset the authors have.

3. R2 is missing for Table S7, and p-values are only provided for the GLMM between total genetic load to LMS. There is no information about whether the p-values have been adjusted for multiple testing either. I understand that p-values are affected by many factors especially in less straightforward tests like GLMM, and $p < 0.05$ is not a gold standard. However, I think it is still necessary to provide p-values (post adjusting of course) as it is a relatively easy way to understand the significance of the test, especially when most R2 values are very close to zero, meaning genetic load can only explain 5%-7% of the fitness. Extended data Figure 1 also showed that the variances of genetic load estimates are big I am not an expert of statistics, so if there are other better ways to represent the significance, I'd love to learn about them. Overall, provided results of the statistical analyses is not sufficient to lead to the conclusions. Therefore, I'm not able to judge whether the main statements are correct.

4. Overall, the testing of GERP score fit more of the authors' expectations than SnpEff results. However in Figure 2C, GERP predicted genetic load in exons showed a positive correlation with LMS. The authors stated that it "potentially reflecting signals of ongoing adaptation or functional turnover, which can affect the relationship between GERP scores and the strength of selection". However, this contradicts to the basic assumption that derived alleles with high GERP scores are deleterious. The statement works only when most derived alleles in exons with high GERP scores are adaptive while most equivalent alleles not in exons are deleterious, which would not be a reasonable assumption to make given the dominance of purifying selection in the genome, especially in conserved genes.

5. It is indeed always very difficult to link genetic features to complex life history traits, such as LMS, as too many factors are coming into effect. With this comprehensive dataset, I have a few suggestions hopefully can help to improve the manuscript: A. instead of trying to find correlations with all individuals, which has introduced massive noise, maybe try to compare individuals with very high/low LMS. B. Try to locate the region(s) where genetic load has the most effect on fitness. The random sampling test of deleterious mutations (extended data Figure 3) showed wide distributions, and it is likely that deleterious mutations in certain regions play a stronger role in fitness. If not find any significant region, that can also be explained by complex effect related to overall physical conditions.

Below are minor comments:

Line 36: the authors probably mean "homozygous and heterozygous deleterious mutations"?

Line 49-50: The most common form of increased genetic load is the exposure of recessive deleterious alleles in homozygous form. I believe the authors fully understand this concept, but the expression may confuse readers to think about accumulation of new mutations.

Line 50-51: I don't understand what the phrase "induced mutations" refers to.

Line 53: I can't agree that the sex difference of genetic load is well studied enough to say that males are more affected. The phrasing might need to be adjusted.

Line 77-80: "This comprises homozygous recessive deleterious mutations, which are expected to have large effect sizes but be relatively infrequent, and heterozygous partially recessive mutations, which should have smaller effect sizes but be more numerous due to the reduced efficacy of purifying selection against them." I wonder from what studies this very detailed information come from. As far as I know, neither the frequency of partially recessive deleterious alleles, or the distribution of fitness effects of deleterious alleles (no matter what dominance) is still not well studied. Even in the cited article, the authors discussed that "we still lack information on the relationship between the h or s coefficient, and the genetic architecture of a trait".

Line 116: The term "genetic load" essentially refers to the negative fitness effect. I'd suggest to use either "quantify genomic mutation load" or "quantify the fitness effects of deleterious alleles".

Line 123-124: It is interesting to see reporting the total number of reads, which is uncommon.

Line 125: In the supplementary the authors wrote "27 full siblings (5.8%), 60 parent-offspring 32 pairs (12.9%), 176 second-degree relatives (37.8%) and 203 third-degree relatives (43.6%)" which I don't think it's sufficient to say that the dataset "primarily consisted of unrelated individuals". While I don't think this affects the main analyses and results, the authors should still make a clear statement in the main manuscript and explain why the relatedness between samples does not affect their main conclusions.

Line 172-174: The authors did not mention whether the sites with flagged with warning messages from SnpEff were filtered

for all analyses, and thus I'd assume they were not. This is worrying because only SNPs with high quality should be used for genetic load analyses. Depending on what warning messages were delivered, some warnings may suggest the quality of the reference genome or annotation is not high, which can be also worrying.

Line 411: Just out of curiosity, how strong is the correlation between the number of offsprings and lifetime mating success, which I assume is well studied in many species?

Reviewer #2 (Remarks on code availability):

I have briefly gone through the codes and found the codes are comprehensive and very well organized. The README is very clear and it is very easy to locate scripts for every analysis. I was not able to go through every script but the few scripts I checked look functional and very well coded.

Reviewer #3 (Remarks to the Author):

Predicted deleterious mutations reveal the genomic mechanisms underlying fitness variation in a lekking bird

This manuscript addresses a fundamental question in evolutionary and conservation biology: Does deleterious genetic variation that is continuously created by mutation impact fitness in a non-isolated population of a widespread species with very strong sexual selection? To answer this question, this manuscript combines estimates of genomic load, inferred from 190 whole-genome sequences from male black grouse from five study sites in Finland, with detailed life-history data spanning a decade. Together with a broad coverage of the literature, this results in a very interesting contribution to an important area of evolutionary research.

I very much enjoyed reading the paper. I particularly enjoyed the fact that the heterozygous load was explicitly included in the analyses. As long ago as Muller (1950) and Morton, Crow & Muller (1956), theory predicted that deleterious mutations would have their biggest impact through the heterozygotes, because most deleterious mutations occur at a low frequency and, thus, are typically carried in heterozygous genotypes. The explicit analysis of the heterozygous load is a great strength of this manuscript, and it deserves the attention it receives in the conclusions.

I also appreciate the clean, easy to follow code and the detailed readme files on GitHub.

Below are some comments that will hopefully help in the revision of the manuscript. I'll begin with some general comments, followed by more specific comments:

1) The main focus of this manuscript is the calculation of genomic load, and its association with fitness variation. Given that focus, I missed a more detailed presentation of these estimates of genetic load. I didn't find a figure or a table that showed the frequency distribution of estimates of load (total, homozygous, and heterozygous) across the 190 individuals, or the correlations between the different types of load among individuals. Figure 1 gives information about the number of SNPs that fall into the different categories of predicted deleteriousness, and Extended Data Figure 1 plots lifetime mating success against the load. Both these figures are very helpful. Given the central role that estimates of genomic load play in this manuscript, I feel that an additional figure showing the main features of the load data should be part of the manuscript.

2) The authors identify more than 400'000 SNPs that fall into the highest GERP score category (here: ≥ 4). This is two orders of magnitude more than what typically has been found in humans, other mammals, and birds. This huge discrepancy should be explored and discussed. What explains this large number? Is it really biologically plausible that 400'000 SNPs with the highest predicted deleterious effects segregate in black grouse? Also, individuals were found to carry more than 120'000 highly deleterious mutations on average (line 157). If we assume - in a back of the envelope kind of way - that black grouse have approximately 20'000 genes, this would imply that - on average - every gene contains 6 highly deleterious SNPs. Is this plausible?

One possible explanation is that GERP scores are good at separating neutral alleles from alleles under selection, but less effective at discriminating weakly from highly deleterious mutations (see Grossen et al. 2020, <https://doi.org/10.1038/s41467-020-14803-1>, for such a finding). Figure 1a hints that perhaps something similar might be happening in black grouse. The four highest GERP score categories all contain roughly similar number of SNPs. Taken at face value this implies that highly deleterious and weakly deleterious SNPs are roughly equally frequent, which seems biologically not very plausible. An alternative explanation would be that GERP scores cannot distinguish well between different levels of deleteriousness. This may be worth exploring in more detail, also in the literature.

3) Inbreeding is known to occur in this population of black grouse, and inbreeding depression in lifetime mating success has also been demonstrated (Hoglund et al. 2002). Thus, it seems plausible that at least the effects of the homozygous and the total load on lifetime mating success represent general inbreeding depression, not necessarily only caused by the loci used in the calculations of the load. Other studies have found strong correlations between estimates of genomic load and estimates of individual inbreeding (e.g. Taylor et al. 2024, <https://doi.org/10.1016/j.cub.2024.02.002>). This manuscript would gain substantially from a similar analysis and a discussion of the possibility that some measures of load simply pick up variation in genome-wide homozygosity caused by inbreeding, just because homozygosity of SNPs correlates with inbreeding and not because one was successful at identifying functionally important SNPs. If measures of genomic load and inbreeding correlate in black grouse, then the association between fitness and measures of genomic load does not

necessarily validate the successful detection of deleterious SNPs. Of course, this does not apply to the heterozygous load. The relationship between measures of load and inbreeding would be an important aspect to cover in this paper.

Specific comments:

- Title: I am not sure that this paper really reveals the genomic mechanisms underlying fitness variation. The paper reveals an association between estimates of genomic load and lifetime mating success, but the genomic mechanisms (number of loci, interactions within (dominance) and among loci, role of the genes affected, etc.) are not part of this manuscript. Thus, I think a title that has the reduction of lifetime mating success through genomic load as its focus would tell readers more precisely what to expect in this paper.

- L. 35: What do the authors mean when they write that deleterious mutations are "strongly" associated with reduced male lifetime mating success? The R^2 of the underlying statistical models are very low (Suppl. Tables 1-3), so the associations aren't strong in the sense that genomic load explains a lot of the variation in fitness (which one wouldn't expect in the first place). Thus, the authors must have another metric or another comparison in mind. It would help readers if the basis for this comparison was made clear.

- L. 36-38 and L. 77-80: These sentences imply that fully recessive deleterious alleles occur more often in homozygous state, while partially recessive deleterious mutations occur more often in heterozygous state. I am not aware of theory or empirical data that show this. What is true, however, is that one observes relationships between the selection coefficient, s , the dominance coefficient, h , and the allele frequencies such that more detrimental mutations are more recessive and rarer than less detrimental mutations. Perhaps what the authors wanted to say in these two places is that fitness is not only reduced through homozygous mutations (as has sometimes been assumed in the early genomic load literature) but also through mutations in heterozygous state, since most deleterious mutations are partially and not fully recessive, even lethals (Morton, Crow & Muller 1956, <https://www.pnas.org/doi/pdf/10.1073/pnas.42.11.855> quote a dominance of lethals and semi-lethals in *Drosophila* of 4-5%).

- L. 43-44: I couldn't find a lot of information about the genetic architecture of sexually selected traits in the manuscript. Perhaps omit this here and on line 379?

- L. 111: Not necessarily here, but somewhere in the manuscript there should be a discussion of how good a measure of fitness (lifetime reproductive success) lifetime mating success is. In black grouse, mating success of males seems to be a good measure of genetic parentage, but lifetime reproductive success depends on a number of other life-history traits (clutch size, hatching success, post-hatching survival, etc.) that may not be captured by lifetime mating success. A discussion of potential caveats that arise from this should be included in the paper.

- L. 160-162: That total GERP and SnpEff loads are not strongly correlated ($r=0.13$) might seem surprising at first sight. Does this imply that GERP and SnpEff measure complementary aspects of the deleteriousness of mutations, in light of the fact that both loads reduced lifetime mating success? Also, I was wondering whether at the SNP level (rather than the load level) GERP and SnpEff scores are related? To that end it would be interesting to add a panel to Figure 1 which shows some summary statistics (the average?) of the GERP scores for each of the four categories of SnpEff.

- L. 164-168, Bayesian GLMMs: A few aspects of these models did not become clear to me. First, I missed an exploration of the sensitivity of these models to the choice of prior, an important element of all Bayesian analyses. Second, I missed the complete model results, including estimates of all the fixed and random effects in the models. Please provide these in the supplementary material. (These first two points also apply to the other Bayesian models in the manuscript). Third, I was wondering whether a random effect for year-of-birth of the males should be included in the models here, as was done for AMS? In the datasets that I am familiar with, fitness measures vary greatly over time because of environmental effects. If one doesn't account for such environmental effects, differences in fitness become difficult to interpret. Lastly, I was curious why the authors ran separate models for GERP and SnpEff loads. Given that they are not strongly correlated and that they might be measuring complementary aspects of load, would it not make sense to include them both in the same model?

- L. 182-185: True but the confidence intervals overlap greatly, so that it is difficult to know how much of the effect is due to the GERP score as opposed to stochastic sampling. An important extension here would be to run the models with all loci with GERP score <0 . This should show a clear and significant difference to the analyses with GERP score ≥ 4 .

- L. 209: Empirical data also support this expectation, see e.g. Di & Lohmueller 2024, <https://doi.org/10.1093/gbe/evae147>.

- L. 267: The text suggests that for SnpEff 1000 mutations were randomly sampled, while Extended Data Figure 3 mentions 5000. Please clarify.

- L. 297-299: It would be good to also consider effect sizes here and not only the overlap of the credible intervals with zero. The effect size of SnpEff load on annual mating success (-0.11) was identical to that on lifetime mating success (-0.11 , Suppl. Table 1)). Thus, for SnpEff load, the estimated impact on AMS and LMS is identical.

- L. 337-345: You could mention here that the importance of heterozygotes was predicted as early as Morton, Crow & Muller 1956 (p. 860: "... most elimination of deleterious genes is now in heterozygotes rather than in homozygotes.)

- L. 357: Given that only GERP load but not SnpEff load affected lek attendance, I think "compelling" is too strong a word here. I'd say that the GERP load results suggest a possible pathway by which deleterious mutations affect mating success, but if the case were clear, SnpEff load would have had to yield similar results.
- L. 381: Measures of genetic and genomic load contain little information about population fitness. See Kardos et al. (<https://doi.org/10.1111/mec.17608>) for a recent review.
- L. 427-430: Please explain what protocol was used for the library preparations and whether it included a PCR step or not.
- L. 433: 0.1% of a read (150bp) is less than a single base. Is this a typo?
- L. 444: What does "coverage" mean here? Average sequencing coverage was 32, so "coverage" must have a different meaning here.
- L. 481-482: Why was the Z but not the W chromosome removed from the analyses?
- Figure 1: I struggled with the colour scheme of Figure 1c because I couldn't differentiate the colours very well. Please use a colour palette that differs more (remember that 8% of men are colour blind...). Also, please add the SFS of neutral SNP to panels d and e. That way readers can see the difference in SFS between neutral and deleterious loci. Finally, note the typo in the legend of panel f ("Homozygoys").
- Extended Data Figure 1 and Supplementary Table 1: A striking result here is that homozygous and heterozygous GERP load each have a stronger association with LMS (beta is more negative by more than a factor 2) than total GERP load. This is not what one would expect if homozygous and heterozygous loads measure different aspects of the load. Is this result partly explained by a negative correlation between homozygous and heterozygous GERP load, so that the variation in total load is not the sum of the variation in homozygous and heterozygous load? This peculiar result should be explored in more detail.

Lukas Keller

Reviewer #3 (Remarks on code availability):

I reviewed the code but I did not install and run it myself.

The code was very clean and easy to follow. The readme files are clear and contain enough information so that I could find all the relevant pieces of code quickly.

*****END*****

Version 1:

Decision Letter:

22nd April 2025

Dear Rebecca,

Your revised manuscript entitled "Genetic architecture of male reproductive success in a lekking bird: insights from predicted deleterious mutations" has now been seen by the same reviewers, whose comments are attached. The reviewers believe the study is improved but still have a few issues which will need to be addressed before we can offer publication in Nature Ecology & Evolution. We will therefore need to see your responses to the criticisms raised and to some editorial concerns, along with a revised manuscript, before we can reach a final decision regarding publication.

We therefore invite you to revise your manuscript taking into account all reviewer and editor comments. Please highlight all changes in the manuscript text file in Microsoft Word format.

* Include a "Response to reviewers" document detailing, point-by-point, how you addressed each reviewer comment. If no action was taken to address a point, you must provide a compelling argument. This response will be sent back to the

reviewers along with the revised manuscript.

* If you have not done so already please begin to revise your manuscript so that it conforms to our Article format instructions at <http://www.nature.com/natecolevol/info/final-submission>. Refer also to any guidelines provided in this letter.

* Extended Data Figures - please ensure that any supplementary figures and tables that are crucial to the manuscript's conclusions are converted into Extended Data figures and tables to increase visibility of these data. Extended Data figures and tables are online-only (present in the online PDF and full-text HTML versions of the paper), peer-reviewed display items that provide essential background to the article but are not included in the main article due to space constraints. A maximum of ten Extended Data display items (figures and tables) is permitted.

Link Redacted

Nature Ecology & Evolution is committed to improving transparency in authorship. As part of our efforts in this direction, we are now requesting that all authors identified as 'corresponding author' on published papers create and link their Open Researcher and Contributor Identifier (ORCID) with their account on the Manuscript Tracking System (MTS), prior to acceptance. ORCID helps the scientific community achieve unambiguous attribution of all scholarly contributions. You can create and link your ORCID from the home page of the MTS by clicking on 'Modify my Springer Nature account'. For more information please visit www.springernature.com/orcid.

[redacted]

Reviewers' comments:

Reviewer #1 (Remarks to the Author):

I would like to thank the authors of doing an good job adressing the comments by all reviewers and clarifying most of our concerns and still consider this a highly interesting and relevant paper. I just have a few smaller comments:

1. The FROH estimates are - as far as I can understand based on all sizes of ROH. I still find the demographic history of the study population a bit unclear, did you look into the distribution of recent vs more ancient inbreeding signatures? Since this can influence the distribution of mutational load in the population, this might be an aspect worth considering.
2. I am a bit puzzled by the use of lifetime mating success as being directly linked to fitness. As I understand, this is based on the total number of observed matings. To what extent has it been verified that LMS is correlated to LRS, i.e. can matings occur without any chicks being produced? I totally agree about this being an important trait, but I feel like the link to fitness (survival and reproduction) could benefit from further clarificaion.
3. Has inbreeding avoidance been documented in this species?
4. The discrepancies between the GERP and SnpEff has been adressed by the authors, but I would like the authors to motivate the inclusion of both estimates given the strong divergence in the number of mutations identified.

Reviewer #2 (Remarks to the Author):

The authors have done great work addressing the comments and have significantly improved the manuscript. I only have one major concern regarding the GO analysis. The authors selected six GO terms and only did analyses for genes belonging to these GO terms. I'm afraid that it is not an appropriate way to "hand pick" GO terms for functional analyses. I agree that the biological functions of these GO terms are theoretically important, but so as many others. For

example, why choose “androgen metabolic process”, but not the parent term “hormone metabolic process”, as hormone metabolism in general is associated with behavior, body growth, immunity level and so on. My point here is that a functional enrichment analysis is more appropriate to find key functional terms, as reproduction is a complex process, expected to be associated with many major functional pathways. I'd expect genes involved in biological processes important for reproductive success are less likely to harbor deleterious alleles and thus should be under-represented. Besides R packages like topGO, there are also online tools for GO enrichment analysis which might be handy, for example: <https://biit.cs.ut.ee/gprofiler/gost>

Here are several minor comments and replies to the replies by the order of line numbers:

Line 47-48 (comment 18): I think the current expression is clear and precise. However the content in the brackets “(the reduction in fitness due to deleterious mutations)” is repetitive and can be deleted.

Line 49 (comment 19): Thanks for the kind explanation, but what I didn't understand here is that, why it is mentioned here as the effect of de novo (new) mutations is not part of this study. Maybe the authors are trying to refer here is that sexual traits can be affected by deleterious mutations? Maybe consider rephrasing in a way, for example “... and experimental studies showed that sexual traits can be disrupted by deleterious mutations. However, key attributes ... remain poorly understood in wild populations”

Line 85-87: I'd suggest being cautious to state that estimating genetic load is a better way than estimating inbreeding. There's still not a good way to really estimate fitness effects of the “deleterious mutations” in wild populations. And there are studies showing correlations between fitness and inbreeding (e.g. DOI: 10.1016/j.cub.2019.06.064). What is the best way to assess fitness is still very much under debate.

Line 132 (comment 7): I think what Reviewer 1 meant is the coverage (depth) range for the individuals, rather than for the SNPs. It is necessary to show that no sample with low depth is included because for example, for an individual with an average depth of 8x, a minimum SNP depth of 20 would only keep SNPs in regions with depth higher than twice of the genome-wide average, likely repeated regions.

Line 136 (comment 24): The number of related pairs does not reflect the number of individuals involved. Again I don't think relatedness in the samples affects the main results, but I'd suggest providing additional information on how many of the 190 individuals are found to be related to each other, rather than only a slightly misleading number of the proportion of the pairs.

Line 200 (comment 13): Thank you very much for the detailed explanation. I agree that the Bayesian framework applied in this study is appropriate. What I meant initially was to provide p values converted from the standardized beta just to make the results easier to follow, but you did make a point that providing the standardized beta is still the best way. I'd suggest having a very brief statement on how to interpret the results somewhere in this session or in the Methods, just to guide readers like me. For example, from your response: “we consider an effect to be “statistically significant” if its 95% CI does not overlap zero”

Line 301-312 (comment 14): Thanks for sharing the preliminary results. I assume these are from ongoing works. This is very exciting and looking forward to your further studies. However, stating there are unpublished results without giving proof is odd. I think it is enough to mention that one possible explanation is that highly deleterious mutations are selected at the early stage of life, and further study is needed.

Line 802: There is a “standardized” while the rest are “standardised”.

Reviewer #2 (Remarks on code availability):

The codes are comprehensive and very well organized. The README is very clear and it is very easy to locate scripts for every analysis. I have not further checked the codes in this round of review.

Reviewer #3 (Remarks to the Author):

Genetic architecture of male reproductive success in a lekking bird: insights from predicted deleterious mutations

I greatly appreciate the effort the authors have put into the revision of their manuscript, and into explaining their decisions in detail. To my mind, the result is a very nice contribution to the field.

There is only one issue that I think the authors have not dealt with convincingly. It concerns the very large number of SNPs that are categorised as deleterious by both GERP and SnpEff (comment 28 in the rebuttal letter). Here is why I think the issue should be addressed head on in the manuscript:

I agree that comparing the absolute number of deleterious SNPs is not straightforward for a number of reasons which the authors highlight in their rebuttal letter. Nevertheless, the absolute number of deleterious alleles is what determines the genetic load (lines 614-616 in the revised manuscript), and hence the absolute number of deleterious SNPs is an essential quantity that cannot be ignored.

The authors argue that other papers have also found such high numbers of deleterious SNPs. While one study of wolves has done so, some studies of birds have reported a much smaller number. For example, Mather et al. (2023, Evolution) have found ~12'700 deleterious SNPs among 66 sequenced Montezuma Quails with a total of 9.3 Mio. SNPs. This corresponds to a percentage of deleterious SNPs of approx. 0.14%. Even accounting for the differences in sample size, this is nearly ten times less than what the current paper reports.

To my mind, the value of the present study is undisputed. The very high number of predicted deleterious alleles does not distract from that. It simply highlights the fact that it still isn't straightforward to identify deleterious SNPs with a high level of confidence, especially in non-model organisms where annotation and mapping errors are normal. To my mind, all that is needed is a short paragraph highlighting the comparatively high number of deleterious SNPs discovered in this study, and a brief exploration of the possible reasons for it. Annotation and sequencing errors (see reference 49 in the manuscript) would be obvious candidates, but some of the reasons given by the authors in their rebuttal letter might also contribute. All in all, a short paragraph highlighting the difficulty of identifying deleterious SNPs will strengthen and not weaken the impact of this manuscript.

Lukas Keller

*****END*****

Version 2:

Decision Letter:

23rd May 2025

Dear Rebecca,

Thank you for submitting your revised manuscript "Genetic architecture of male reproductive success in a lekking bird: insights from predicted deleterious mutations" (NATECOLEVOL-24113381B). It has now been seen again by Reviewer #2 and their comments are below. The reviewer is satisfied with the revisions and therefore we'll be happy in principle to publish it in Nature Ecology & Evolution, pending minor revisions to comply with our editorial and formatting guidelines.

[redacted]

Reviewer #2 (Remarks to the Author):

Thanks for the great work. I have no further comments.

Reviewers' comments:

Reviewer #1 (Remarks to the Author):

Chen et al. present a highly interesting manuscript and contribute with valuable empirical data to a field that yet remains to be explored. I really enjoyed reading this manuscript and found it well-written and sound, and I was really impressed by the quality of the data set. However, the manuscript feels somewhat overloaded with content that needs to be put in a more extensive theoretical background, so a few changes would hopefully make it even more interesting. These are my main comments:

We thank the reviewer for the positive feedback on our dataset and manuscript, as well as for the suggestion to expand upon the theoretical background, which we have now done (see responses to comments 1 and 34).

Comment 1

- regarding (iii) elucidate the pathways by which deleterious mutations influence reproductive success, focusing on their effects on the expression of behavioural and ornamental traits., I feel like this part is lacking in theoretical context in the introduction, even though it is quite extensively discussed. This part would benefit from clearer theoretical predictions to make it feel less descriptive.

Thank you for pointing out the lack of theoretical context about the sexual traits. To address this comment, we have included a new paragraph in the results section explaining how sexual traits are hypothesized to be condition-dependent, as proposed by the genetic capture hypothesis. Condition is a complex, polygenic trait with a large mutational target, making condition-dependent sexual traits particularly susceptible to the effects of genome-wide mutations (i.e. individual genomic mutation loads). Due to space limitations in the introduction, we expanded upon the theoretical context in a new paragraph in the results / discussion section (lines 340-349).

Comment 2

- the manuscript mentions the value of this study for conservation. However, the applied link needs to be clarified or removed - the value of understanding these links is unquestionable, but the part about optimising conservation strategies and improving captive breeding needs to be more thoroughly explained, or consider removing it.

Thank you for this suggestion. We decided to clarify the applied value for conservation by adding a short paragraph to the introduction (lines 67–75).

Comment 3

- on that note, what is known about the inbreeding level in the study population? There is indeed data to explore ROH and testing if the mutational load increases with inbreeding levels is highly relevant for this study.

Thank you for the suggestion to add information on the level of inbreeding and its relationship to the mutational load in our study population. In response to this

comment and several others (comments 11 and 30), we have now added several new analyses relating to inbreeding.

First, we quantified variation in individual genomic inbreeding coefficients (F_{ROH}), showing that there is appreciable variation within the study population. To address this comment, we wrote a new introductory paragraph on inbreeding (lines 77–87), a new methods paragraph (lines 526–539) and a new results paragraph (lines 139–147). We also produced a new (main) figure (Figure 1).

Second, as requested, we explored the relationship between inbreeding and the various load components. We found that more inbred individuals have higher homozygous loads but there is no association between F_{ROH} and the total load. To address this comment, we wrote a new results paragraph (lines 227–238) and produced a new figure (Extended Data Figure 2).

Comment 4

- reference genome: if I understand correctly, all processing has been done with a black grouse reference genome.

Where is this individual from?

The reference genome that was assembled by the 10K bird project came from a black grouse from Nordland in Norway, which was preserved at the Natural History Museum of Denmark. Black grouse from Norway and Finland are genetically the same subspecies². We have added this information to the Supplementary Methods (lines 73–75).

Comment 5

Can the selection of a con-specific reference genome introduce a bias in mutational classification?

No, we have no reason to believe that using a black grouse reference genome from a closely related population would introduce a bias in mutational classification. First, for the GERP analysis, we used the default setting, which ensures that the focal reference genome is not included in the computation of the GERP scores. Hence, the scores are assigned independently of the chosen reference genome. Second, SnpEff assigns impact classes independently of the reference genome / derived-ancestral state, as the impact class is only dependent on the location of the mutation and the type of substitution.

Comment 6

- exploring the gene ontology could be important to understand the homozygous and heterozygous load and the link to LMS. Had that been done?

Thank you for this suggestion. We have now added an analysis that utilises gene ontology (GO) to evaluate whether deleterious mutations that affect certain biological processes reduce LMS. First, we hypothesized that deleterious mutations affecting six candidate biological processes (androgen metabolism, cellular respiration,

developmental growth, immune function, muscle tissue development and responses to oxidative stress) could be important for explaining variation in male reproductive success in the black grouse (see Supplementary Table 5 for detailed [referenced] descriptions of our hypotheses and their rationale). To test this, we used GO annotations to extract the deleterious mutations located in the genes associated with each biological process. We then computed the total load separately for each GO term and prediction approach, separately for promoters, introns and exons, and used the resulting values as predictor variables in separate Bayesian GLMMs of LMS.

Three distinct patterns emerged:

- (i) For all six GO terms, GERP mutations in promoter regions were negatively associated with LMS.
- (ii) For three GO terms (androgen metabolism, immune response and oxidative stress response), GERP mutations were negatively associated with LMS in more than one genomic region.
- (iii) For the GO term immune response, SnpEff mutations in all genomic regions were negatively associated with LMS.

Our results again indicate that GERP mutations in regulatory regions have large phenotypic effects and point towards important roles of androgens, immunity and oxidative immune genes in black grouse reproduction.

We have edited the introduction accordingly (lines 96–103, 125–127), and added new paragraphs to the methods (667–683) and results (lines 314–338), as well as two new Extended Data Figures (Extended Data Fig. 5–6) and four new Supplementary Tables (Supplementary Tables 5–8).

Comment 7

- L123: what range of coverage? Did you exclude any due to low coverage?

The range of the mean coverage across individuals per raw (unfiltered) SNP is 0.005 – 368X. We excluded SNPs with a minimum depth of coverage below 20X and a maximum depth of coverage greater than twice the mean depth (60X). This additional information has been added to the genotyping methods (lines 512-513).

Comment 8

- L386 and onwards: the use of core males is motivated to explore the sexual selection perspective, however an important part of the story goes missing without the females. Do you have samples from them? What do you know about their mutational load?

We agree that studying the mutation load in females would be fascinating, and one of us (C.S.) does indeed have some samples. However, at the moment, we unfortunately lack the resources to sequence and analyse female black grouse genomes. To provide context, this paper represents the main outcome of the PhD project of Rebecca Chen, which focuses predominantly on sexual selection in male black grouse. Rebecca is

currently in the final year of her PhD. We do, however, have plans to write a follow-up grant to the German Science Foundation once this project is complete, as learning more about the mutation load of the females would provide valuable additional context and insights into the mutation load dynamics of this species.

Comment 9

- is the study population temporally stable or fluctuating?

The study population has fluctuated throughout our study period. Specifically, while in earlier years (2002 to 2004) the population was relatively stable, population density increased between 2005 and 2007 and remained stable from 2007 to 2008³. To account for these fluctuations, which might impact male-male competition, female numbers and mating success, we included year and lek site as random effects in our models. We have now explicitly explained how our modelling framework accounts for population fluctuations in lines 360–361.

Reviewer #2 (Remarks to the Author):

This study has a great dataset of life-long fitness estimates which is very valuable for understanding genetic load. Bioinformatic works are very well done. However, the results of statistical tests are not sufficient to reach the major goal of linking genetic load to life-time reproductive success and the contradiction in the main results is unneglectable. Please find major comments below:

Thank you for these kind words about our dataset and the quality of the bioinformatic methods. As described below, we have used Bayesian mixed models to analyse our data, which do not rely on arbitrary cut-offs (for example p-values) for statistical inference, but rather they rely on parameter estimation. We hope the arguments and additional explanations we provide in our detailed responses to comment 13 will help to convince the reviewer of the statistical rigour of our analyses.

We interpret the contradiction in the main results that the reviewer is referring to as the unexpected finding that total GERP load in exonic regions is positively associated with LMS. At the suggestion of the reviewer (see responses to comment 14), we delved into this in greater detail. Based on a preliminary analysis of a small unpublished dataset of chick genomes, we believe we now have a clear explanation for this unexpected result (see response 14 for details).

Comment 10

1. The structure of Introduction can be improved so that the readers can follow more easily. Now the Introduction starts with the fitness effects of genetic load (or mutation load), then goes to technical prospect of measuring genetic load with genomic data, and then goes back to the genetic principle of genetic load. I'd suggest the Introduction to be organized following the order of theory (how genetic load works and linkage to sexual selection) – technic (genomics) – study case (black grouse, why it is suitable, and research questions).

Thank you for this suggestion. The introduction starts with what we know about the fitness effects of the genetic load *based on non-genomic data*, which we have now clarified. In the proposed restructure, we were unsure where the paragraphs on the genetic architecture of fitness (e.g. covering the roles of homozygous versus heterozygous mutations and coding versus non-coding mutations) should go, as these paragraphs contain both theoretical expectations about deleterious mutations as well as knowledge of the methods used to identify mutations. However we have now extensively restructured and reorganised our introduction to incorporate additional referee comments as follows: (i) we added a paragraph that explains why this research is important for conservation (see response to comment 2); (ii) we added a paragraph to contextualise the new inbreeding analyses by reference to the expected fitness effects of the genomic mutation load ; and (iii) we clarified what we mean by genetic architecture. We hope the overall structure of the introduction is now clearer.

Comment 11

2. I'd suggest to measure and present genomic heterozygosity. It is a basic measurement reflecting "genetic quality" of an individual. The connection between genetic diversity (often measured as heterozygosity), genetic load and fitness has been widely discussed in population genetic and conservation genetics. It would be interesting to have a deeper look with such good dataset the authors have.

Thank you for this suggestion. Genomic heterozygosity as a measure of individual quality is tightly linked (and essentially the reverse of) genomic inbreeding. As the suggestion to include inbreeding in the paper was also made by the other reviewers (see comments 3 and 30), we have now added analyses of genomic inbreeding to the manuscript. To address the comment about the connections between genetic diversity, the mutation load and fitness, we implemented several new analyses. Specifically, we investigated the relationships between F_{ROH} , the total load and the homozygous and heterozygous loads, and we also tested for inbreeding depression (for details, please see our response to comment 3).

Comment 12

3. R2 is missing for Table S7

There are no R^2 values in Table S7 because the values presented in this table are based on the models presented in revised Tables S9–S11 (which all include R^2 values). The direct and indirect effects presented in Table S7 were calculated using the product method (see Methods for details) where the posterior distributions of multiple models are used to calculate the indirect effect, and therefore R^2 cannot be estimated. We have now explained this in the legend of the table, which is now revised Supplementary Table 12 (lines 306-308).

Comment 13

and p-values are only provided for the GLMM between total genetic load to LMS. There is no information about whether the p-values have been adjusted for multiple testing either. I understand that p-values are affected by many factors especially in less straightforward tests like GLMM, and $p < 0.05$ is not a gold standard. However, I think it is

still necessary to provide p-values (post adjusting of course) as it is a relatively easy way to understand the significance of the test, especially when most R2 values are very close to zero, meaning genetic load can only explain 5%-7% of the fitness. Extended data Figure 1 also showed that the variances of genetic load estimates are big I am not an expert of statistics, so if there are other better ways to represent the significance, I'd love to learn about them. Overall, provided results of the statistical analyses is not sufficient to lead to the conclusions. Therefore, I'm not able to judge whether the main statements are correct.

First, thank you for pointing out that the GLMM comparing the mutation loads of the different lekking sites (Supplementary Table 13) was not implemented in a Bayesian framework. This could indeed be confusing given that all of the other analyses were done using Bayesian statistics. To further harmonise the statistical methodology and reporting of the results in our manuscript, we have therefore re-implemented this analysis in a Bayesian framework.

Unlike frequentist statistics, Bayesian approaches do not rely on p-values for biological inference. Instead, Bayesian inference focuses on parameter estimation⁴, where statistical significance is assessed through probability distributions rather than binary hypothesis testing. While p-values are informative if used correctly, they do not always provide a robust measure of evidence for or against a hypothesis⁵.

We do not wish to argue that one framework is better than the other – they are based on different philosophies. In Bayesian statistics, the parameters are estimated iteratively, with the most frequently observed values considered the most likely to reflect the true parameter value. In essence, we look at the probability of a value (such as the beta estimate of the regression) lying within a certain range. The posterior distribution (e.g. shown in Figure 3a) then shows the distribution of estimates for a specific parameter (here, the beta estimate of the total load).

In Bayesian statistics, credible intervals (CIs) quantify the uncertainty of the estimated values. If the 95% CI of a given value ranges from 0.5 to 1, this means that there is a 95% probability that the true value lies within this range. Conventionally, we consider an effect to be “statistically significant” if its 95% CI does not overlap zero. This is analogous to the use of an alpha value of 0.05 in frequentist statistics. All of the major conclusions drawn in our manuscript follow this convention.

Finally, frequentist statistics rely on controlling for the probability of false positives using significance thresholds. When performing multiple tests, the probability of obtaining at least one false positive increases, requiring Bonferroni or False Discovery Rate correction. However, correcting for multiple testing is not relevant in a Bayesian framework because null hypothesis significance testing is not performed and hence there are no p-values to correct⁶.

For more resources on Bayesian modelling, please see references ^{7,8}.

Furthermore, we would like to point out that our conclusions reached with Bayesian statistics are basically the same as those reached with frequentist statistics. To illustrate this point, we repeated one of our main statistical models (LMS ~ total GERP load) using frequentist statistics. We fitted the otherwise identical regression using the R package glmmTMB with a Poisson distribution and a zero inflation of ~1. As shown in Tables 1 and 2, we obtain very similar beta estimates with the two approaches and the corresponding p-value for the total GERP load is tiny ($1e^{-9}$), indicating that the reported effect is indeed highly significant.

Parameter	Estimate	SE	z-value	p-value
Intercept	1.78	0.19	9.26	$2.00e^{-16}$
Total GERP load	-0.20	0.03	-6.11	$1.00e^{-9}$

Table 1. Frequentist model output for the GLMM of the total GERP load on LMS.

Parameter	Median	95% CI	80% CI
Intercept	1.73	0.91, 2.36	1.31, 2.08
Total GERP load	-0.21	-0.27, -0.14	-0.25, -0.17

Table 2. Bayesian model output for the GLMM of the total GERP load on LMS.

We appreciate the reviewer’s request to include p-values in the manuscript and would like to accommodate it. However, doing so would require duplicating all of our data analyses. In practice, this would mean adding over 40 frequentist models together with additional methods, descriptions of the results, and model outputs in the supplementary information. Given that the Bayesian and frequentist models yield essentially the same results (see above), our main conclusions would remain unchanged. To avoid redundant statistical analyses and to keep the length of the manuscript and supplementary materials within reasonable bounds, we would therefore prefer to not include frequentist models (and hence p-values).

Comment 14

4. Overall, the testing of GERP score fit more of the authors’ expectations than SnpEff results. However in Figure 2C, GERP predicted genetic load in exons showed a positive correlation with LMS. The authors stated that it “potentially reflecting signals of ongoing adaptation or functional turnover, which can affect the relationship between GERP scores and the strength of selection”. However, this contradicts to the basic assumption that derived alleles with high GERP scores are deleterious. The statement works only when most derived alleles in exons with high GERP scores are adaptive while most equivalent alleles not in exons are deleterious, which would not be a reasonable assumption to make given the dominance of purifying selection in the genome, especially in conserved genes.

Indeed, this was an unexpected finding. To investigate further, we conducted a preliminary analysis of whole genomes from 46 black grouse chicks. We found that the

chicks have a higher total GERP load than adults, but only at exons (see Figure 1 below). This suggests that many GERP mutations in exons reduce early-life survival and are therefore purged from the population prior to adulthood. Consequently, exonic GERP mutations persisting in the adult population may be less harmful or even beneficial due to ongoing adaptation⁹ or functional turnover¹⁰, which may explain the unexpected net-positive relationship between total exonic GERP load and LMS. We have now expanded upon this point and provided a more detailed explanation in lines 301-312 of the results.

Figure 1. Differences in the total load of (male) chicks and adult male black grouse. Shown are the differences in the total GERP load (left) and the total SnpEff load (right) calculated separately for exons, introns and promoters (top to bottom, respectively). The thick horizontal lines indicate mean total load values, while the lower and upper hinges correspond to the first and third quartiles, respectively, and the whiskers represent 1.5 times the interquartile range. An Analysis of Variance (ANOVA) was used to test for differences in the total load between chicks and adults, with p-values in each panel indicating statistical significance.

Comment 15

5. It is indeed always very difficult to link genetic features to complex life history traits, such as LMS, as too many factors are coming into effect. With this comprehensive dataset, I have a few suggestions hopefully can help to improve the manuscript:
A. instead of trying to find correlations with all individuals, which has introduced massive noise, maybe try to compare individuals with very high/low LMS.

Thank you for these suggestions. However, we are unclear about what is meant by “massive noise” in our data. In the black grouse, a small number of top males secure the majority of matings, while many individuals do not reproduce at all. This variation is a genuine biological phenomenon that results in the LMS data being non-gaussian, right-tailed and zero inflated. To adjust for this, we used a zero-inflated Poisson model, which is the most appropriate family for our data distribution.

At the suggestion of the reviewer, we converted the LMS counts to a binary variable classifying each male as being either successful ($LMS > 0$, $n = 95$) or unsuccessful ($LMS = 0$, $n = 95$) at mating. Using a Bernoulli distribution for an otherwise identical GLMM, we did not detect any significant associations between the total GERP or the total SnpEff load and LMS (see Figure 2 below, as in both cases the 95% CIs overlap zero). This suggests that simplifying continuous variation in LMS into a binary variable results in the loss of important biological information. While we appreciate the suggestion to compare individuals with high versus low reproductive success, in this particular instance, this approach does not appear to increase the power to detect genetic effects. Therefore, we maintain that retaining all of the variation in LMS and using a zero-inflated Poisson model is the most appropriate approach for our data.

Figure 2. Fitness effects of the total genomic mutation load using a binary classification of male mating success. Shown are the posterior distributions of the standardized β estimates of the total GERP load (top) and the total SnpEff load (bottom) on lifetime mating success, coded as successful ($LMS > 0$) versus unsuccessful ($LMS = 0$). The white circles represent the median posterior estimates, the thick black lines the 80% CIs, and the thin black lines the 95% CIs.

Comment 16

B. Try to locate the region(s) where genetic load has the most effect on fitness. The random sampling test of deleterious mutations (extended data Figure 3) showed wide distributions, and it is likely that deleterious mutations in certain regions play a stronger role in fitness. If not find any significant region, that can also be explained by complex effect related to overall physical conditions.

We appreciate that it would be interesting to locate genomic regions where the mutation load has strong effects on fitness. However, our sample size of individuals is too small to implement a meaningful analysis. Genome wide association studies (GWAS) and related approaches typically rely on thousands to millions of study subjects to robustly test for statistical associations¹¹. With our sample size of 190 individuals, any associations that we would detect would likely be spurious. To put this into context, we are only aware of a single study that attempted to map the strength of inbreeding depression across the genome in Soay sheep¹². This was only possible thanks to the availability of SNP array data for almost 6,000 sheep. Part of the problem is that the vast majority of deleterious mutations are individually rare (see Figure 2d,e), which offers very little power to detect associations between fitness measures and deleterious mutations without sample sizes that are currently unfeasible for studies of wild populations. Given this important limitation, our workaround was to implement a GO analysis that seeks to identify specific biological processes where the mutation load has detrimental effects on fitness (see response to comment 5).

Comment 17

Below are minor comments:

Line 36: the authors probably mean “homozygous and heterozygous deleterious mutations”?

This is correct, we have now clarified this point and added *deleterious* to the abstract

Comment 18

Line 49-50: The most common form of increased genetic load is the exposure of recessive deleterious alleles in homozygous form. I believe the authors fully understand this concept, but the expression may confuse readers to think about accumulation of new mutations.

Thanks for pointing that out. To avoid the potential for confusion, we have now simplified our definition of individual mutation load to “the reduction in fitness due to deleterious mutations” as defined in Bertorelle et al.¹³.

Comment 19

Line 50-51: I don’t understand what the phrase “induced mutations” refers to.

In experimental laboratory studies, mutations can be induced in specific cohorts to estimate the effects of the mutation load on fitness. Mutations can be chemically induced¹⁴ or induced with ionizing radiation¹⁵. We have given an example of this in line 49 to give readers a better understanding of what “induced” means if they are not

familiar with such experiments.

Comment 20

Line 53: I can't agree that the sex difference of genetic load is well studied enough to say that males are more affected. The phrasing might need to be adjusted.

We adjusted the phrasing by removing “especially in males”.

Comment 21

Line 77-80: “This comprises homozygous recessive deleterious mutations, which are expected to have large effect sizes but be relatively infrequent, and heterozygous partially recessive mutations, which should have smaller effect sizes but be more numerous due to the reduced efficacy of purifying selection against them.” I wonder from what studies this very detailed information come from. As far as I know, neither the frequency of partially recessive deleterious alleles, or the distribution of fitness effects of deleterious alleles (no matter what dominance) is still not well studied. Even in the cited article, the authors discussed that “we still lack information on the relationship between the h or s coefficient, and the genetic architecture of a trait”.

We have now removed this sentence from our revised introduction.

Comment 22

Line 116: The term “genetic load” essentially refers to the negative fitness effect. I'd suggest to use either “quantify genomic mutation load” or “quantify the fitness effects of deleterious alleles”.

We agree that in theoretical work, “genetic load” refers to the portion of the mutations that have negative fitness effects. We have now rephrased this sentence to “quantify the fitness effects of predicted deleterious mutations” to avoid confusion (lines 124-125).

Comment 23

Line 123-124: It is interesting to see reporting the total number of reads, which is uncommon

Thank you.

Comment 24

Line 125: In the supplementary the authors wrote “27 full siblings (5.8%), 60 parent-offspring 32 pairs (12.9%), 176 second-degree relatives (37.8%) and 203 third-degree relatives (43.6%)” which I don't think it's sufficient to say that the dataset “primarily consisted of unrelated individuals”. While I don't think this affects the main analyses and results, the authors should still make a clear statement in the main manuscript and explain why the relatedness between samples does not affect their main conclusions.

The result in the supplementary information that led to the conclusion that the dataset “primarily consisted of unrelated individuals” is stated before the quote cited by the

referee: “We identified a total of 466 pairs of close kin among 17,949 pairwise comparisons (2.6%)”. Put another way, this means that, among all pairwise comparisons of individual black grouse males, 97.5% of pairs were classified as unrelated, whereas only 2.6% were classified as related. The subsequent breakdown and its corresponding percentages refer only to the 2.6% of pairwise comparisons that involve related individuals. We have now clarified this point in lines 30-32 of the supplementary information and included the statement requested by the referee in the Results section (lines 133-137).

Comment 25

Line 172-174: The authors did not mention whether the sites with flagged with warning messages from SnpEff were filtered for all analyses, and thus I'd assume they were not. This is worrying because only SNPs with high quality should be used for genetic load analyses. Depending on what warning messages were delivered, some warnings may suggest the quality of the reference genome or annotation is not high, which can be also worrying.

Thank you for pointing out this ambiguity. To clarify: we excluded SnpEff annotations that contained a warning message from all of our analyses. To highlight the importance of this filtering step, we mentioned that including warning messages did not result in a significant negative relationship between total SnpEff load and LMS (lines 209-212). We have now added a line in the methods clarifying that we excluded all SNPs with annotations containing warning messages (lines 588-590).

Comment 26

Line 411: Just out of curiosity, how strong is the correlation between the number of offsprings and lifetime mating success, which I assume is well studied in many species?

Lebigre *et al.*¹⁶ provide strong evidence that mating success reliably reflects male reproductive success in black grouse. In 94% of cases where genetic paternities could be compared to mating observations (based on 340 hatchlings from 48 broods), the presumed father sired all of the hatchlings of the brood. A further 4% of broods were sired by multiple males, including the presumed father, while only 2% of the broods were sired by another male. This high consistency between observed matings and true paternity reflects the species' highly competitive mating system.

Reviewer #2 (Remarks on code availability):

I have briefly gone through the codes and found the codes are comprehensive and very well organized. The README is very clear and it is very easy to locate scripts for every analysis. I was not able to go through every script but the few scripts I checked look functional and very well coded.

Thank you.

Reviewer #3 (Remarks to the Author):

This manuscript addresses a fundamental question in evolutionary and conservation biology: Does deleterious genetic variation that is continuously created by mutation impact fitness in a non-isolated population of a widespread species with very strong sexual selection? To answer this question, this manuscript combines estimates of genomic load, inferred from 190 whole-genome sequences from male black grouse from five study sites in Finland, with detailed life-history data spanning a decade. Together with a broad coverage of the literature, this results in a very interesting contribution to an important area of evolutionary research.

I very much enjoyed reading the paper. I particularly enjoyed the fact that the heterozygous load was explicitly included in the analyses. As long ago as Muller (1950) and Morton, Crow & Muller (1956), theory predicted that deleterious mutations would have their biggest impact through the heterozygotes, because most deleterious mutations occur at a low frequency and, thus, are typically carried in heterozygous genotypes. The explicit analysis of the heterozygous load is a great strength of this manuscript, and it deserves the attention it receives in the conclusions.

Thank you for this positive feedback and your kind words.

I also appreciate the clean, easy to follow code and the detailed readme files on GitHub.

Thank you again.

Below are some comments that will hopefully help in the revision of the manuscript. I'll begin with some general comments, followed by more specific comments:

Comment 27

1) The main focus of this manuscript is the calculation of genomic load, and its association with fitness variation. Given that focus, I missed a more detailed presentation of these estimates of genetic load. I didn't find a figure or a table that showed the frequency distribution of estimates of load (total, homozygous, and heterozygous) across the 190 individuals, or the correlations between the different types of load among individuals. Figure 1 gives information about the number of SNPs that fall into the different categories of predicted deleteriousness, and Extended Data Figure 1 plots lifetime mating success against the load. Both these figures are very helpful. Given the central role that estimates of genomic load play in this manuscript, I feel that an additional figure showing the main features of the load data should be part of the manuscript.

Thank you for this helpful suggestion. We have now revised our main figures and also included a new supplementary figure (Extended Data Figure 2) that provides additional information about the various load estimates (i.e. the total, homozygous and heterozygous loads) and their relationship to genomic inbreeding (see also responses to comments 3 and 11). We hope this addresses your comment and would be happy to incorporate any further feedback.

Comment 28

2) The authors identify more than 400'000 SNPs that fall into the highest GERP score category (here: ≥ 4). This is two orders of magnitude more than what typically has been found in humans, other mammals, and birds. This huge discrepancy should be explored and discussed. What explains this large number? Is it really biologically plausible that 400'000 SNPs with the highest predicted deleterious effects segregate in black grouse? Also, individuals were found to carry more than 120'000 highly deleterious mutations on average (line 157). If we assume - in a back of the envelope kind of way - that black grouse have approximately 20'000 genes, this would imply that - on average - every gene contains 6 highly deleterious SNPs. Is this plausible?

Thank you for pointing out the seemingly high number of “deleterious” mutations identified in the black grouse. However, comparing the absolute number of SNPs that fall within a certain GERP score category between study systems is not straightforward for the following reasons:

(i) Genetic diversity and sample sizes

the total number of deleterious SNPs that can be identified is highly dependent on the total number of SNPs in the focal population (i.e. genetic diversity). This in turn depends on various biological parameters such as the mutation rate, effective population size, demographic history, strength of selection, amount of inbreeding and purging. It furthermore depends on methodological factors such as the sample size, sequencing depth, sequencing method and the applied filtering steps. These complexities make it challenging to make direct cross-species comparisons.

To illustrate this point, our sample size of black grouse genomes ($n = 190$) is larger than the sample sizes typically used in similar studies, where sample sizes often consist of around 20 individuals per population / species. If we randomly select 20 black grouse from our dataset, the number of SNPs annotated with high GERP scores decreases by almost 35% to 270,255 SNPs. This dependence on sample size arises because most GERP mutations are present at very low frequencies.

(ii) Methodological aspects of the GERP analysis

The GERP-score cutoff when a mutation is considered “deleterious” is also to some extent arbitrary and varies widely across studies. Additionally, the range of GERP scores depends on the methodological steps taken to construct the multi-alignment file used for the GERP score calculations. This makes it even more difficult to compare the number of deleterious mutations across studies, as there are several decision-making steps that are not standardised and which influence the absolute number of deleterious mutations identified.

(iii) GERP versus SnpEff

It is also important to note that the GERP analysis is not restricted to gene bodies, meaning that intergenic SNPs with high GERP scores can also be detected. This contributes to the observation that GERP++ detects many more mutations than SnpEff (in our study, we identified 77 times more mutations with GERP than SnpEff), which

primarily focuses on coding regions, and which is the method used by most studies. Hence, the number of identified deleterious mutations depends heavily on the prediction approach, which varies from study to study.

(iv) Literature comparison

Delving deeper into the literature, we could not find many studies reporting raw numbers of deleterious SNPs inferred from whole genome resequencing data in either model or non-model organisms; most studies instead report the estimated load values^{17,18}. The developers of GERP++ reported 214,749,502 constrained nucleotides in humans (7% of the genome¹⁹), though this does not refer to genetic variants. Among the few studies that do report raw counts, we found a study of wolves²⁰ that reported 376,835 deleterious GERP mutations (7% of SNPs), which is similar to our proportion of identified deleterious GERP mutations (413,489/6,954,487 = 5.9%). Another study compared the number of deleterious GERP mutations (defined as a GERP score > 20) across Lepidoptera species²¹, and found that this number ranges from 484 sites to 49,898, further highlighting the heterogeneity of this raw number depending on the species and thresholds set.

For the reasons outlined above, we prefer not to directly interpret the absolute numbers of inferred deleterious mutations in any given study. Given the various biological and methodological complexities and the results of the studies mentioned above, we also do not consider our results to be biologically implausible. However, further studies are needed to develop a better picture of how the number of inferred mutations varies across species in response to various biological and methodological factors.

Comment 29

One possible explanation is that GERP scores are good at separating neutral alleles from alleles under selection, but less effective at discriminating weakly from highly deleterious mutations (see Grossen et al. 2020, <https://doi.org/10.1038/s41467-020-14803-1>, for such a finding). Figure 1a hints that perhaps something similar might be happening in black grouse. The four highest GERP score categories all contain roughly similar number of SNPs. Taken at face value this implies that highly deleterious and weakly deleterious SNPs are roughly equally frequent, which seems biologically not very plausible. An alternative explanation would be that GERP scores cannot distinguish well between different levels of deleteriousness. This may be worth exploring in more detail, also in the literature.

Regarding the performance of GERP for mutations of different effect size classes: this has been extensively discussed and evaluated by Huber et al.²². They show that GERP is less effective at distinguishing weak selection from strong selection, although it performs well at identifying mutations under purifying selection versus those neutrally evolving. This means that mutations with high GERP scores are most likely to be deleterious compared to those with lower or negative GERP scores, but that mutations with the highest GERP scores do not necessarily have the largest effect sizes compared to mutations with lower (but positive) GERP scores. Therefore, mutations with GERP scores ≥ 4 could be moderately to strongly deleterious. We have clarified this point in lines 157-160.

Comment 30

3) Inbreeding is known to occur in this population of black grouse, and inbreeding depression in lifetime mating success has also been demonstrated (Hoglund et al. 2002). Thus, it seem plausible that at least the effects of the homozygous and the total load on lifetime mating success represent general inbreeding depression, not necessarily only caused by the loci used in the calculations of the load. Other studies have found strong correlations between estimates of genomic load and estimates of individual inbreeding (e.g. Taylor et al. 2024, <https://doi.org/10.1016/j.cub.2024.02.002>). This manuscript would gain substantially from a similar analysis and a discussion of the possibility that some measures of load simply pick up variation in genome-wide homozygosity caused by inbreeding, just because homozygosity of SNPs correlates with inbreeding and not because one was successful at identifying functionally important SNPs. If measures of genomic load and inbreeding correlate in black grouse, then the association between fitness and measures of genomic load does not necessarily validate the successful detection of deleterious SNPs. Of course, this does not apply to the heterozygous load. The relationship between measures of load and inbreeding would be an important aspect to cover in this paper.

Thank you for these interesting comments, which align with those of the other reviewers. As described in our responses to comments 3, 11 and 27, we have now investigated the relationships between genomic inbreeding and the various load components, and we also compared the effects sizes of genomic inbreeding and the total load on LRS. We find that F_{ROH} is strongly associated with the homozygous load, but not with the total load. Furthermore, the total load explains more of the variation in LRS than inbreeding. This suggests that both homozygous deleterious mutations, which are expressed to a greater extent in inbred individuals, and heterozygous mutations affect LRS.

Specific comments:

Comment 31

- Title: I am not sure that this paper really reveals the genomic mechanisms underlying fitness variation. The paper reveals an association between estimates of genomic load and lifetime mating success, but the genomic mechanisms (number of loci, interactions within (dominance) and among loci, role of the genes affected, etc.) are not part of this manuscript. Thus, I think a title that has the reduction of lifetime mating success through genomic load as its focus would tell readers more precisely what to expect in this paper.

We acknowledge the reviewer's reservation about the previous title. However, we feel that a more conservative title, such as "predicted deleterious mutations reduce lifetime mating success in male black grouse" would not communicate the breadth of our findings. Our study demonstrates that both homozygous and heterozygous mutations reduce fitness, which sheds some light on dominance effects. We also show that

deleterious mutations in both coding and regulatory regions reduce fitness. Additionally, our new GO analysis suggests that deleterious mutations affect fitness via specific biological processes, including androgen metabolism, oxidative stress responses and immunity (see responses to comment 6). Given the difficulty of encapsulating these insights into a concise title, we aimed to select one that reflects the broader implications of our findings for understanding the genetic architecture of male reproductive success. Consequently, we have revised our title to “Genetic architecture of male reproductive success in a lekking bird: insights from predicted deleterious mutations” and clarified our definition of “genetic architecture” in the introduction.

Comment 32

- L. 35: What do the authors mean when they write that deleterious mutations are "strongly" associated with reduced male lifetime mating success? The R^2 of the underlying statistical models are very low (Suppl. Tables 1-3), so the associations aren't strong in the sense that genomic load explains a lot of the variation in fitness (which one wouldn't expect in the first place). Thus, the authors must have another metric or another comparison in mind. It would help readers if the basis for this comparison was made clear.

We agree that “strongly” is not a particularly helpful term for describing the strength of association. Indeed, the explained variance is not high (as would be expected) although the associated p-values (in a frequentist framework) are very low (see response to comment 13). To address this criticism, we have now removed the word “strongly” from the abstract.

Comment 33

*- L. 36-38 and L. 77-80: These sentences imply that fully recessive deleterious alleles occur more often in homozygous state, while partially recessive deleterious mutations occur more often in heterozygous state. I am not aware of theory or empirical data that show this. What is true, however, is that one observes relationships between the selection coefficient, s , the dominance coefficient, h , and the allele frequencies such that more detrimental mutations are more recessive and rarer than less detrimental mutations. Perhaps what the authors wanted to say in these two places is that fitness is not only reduced through homozygous mutations (as has sometimes been assumed in the early genomic load literature) but also through mutations in heterozygous state, since most deleterious mutations are partially and not fully recessive, even lethals (Morton, Crow & Muller 1956, <https://www.pnas.org/doi/pdf/10.1073/pnas.42.11.855> quote a dominance of lethals and semi-lethals in *Drosophila* of 4-5%).*

We agree this sentence was not very clear. Indeed, we aimed to describe relationship between selection and dominance coefficients in this sentence. We have now reworked our introduction to clarify this point. We now describe how mutation load includes not only homozygous mutations, but also heterozygous mutations, as pointed out by the reviewer.

Comment 34

- L. 43-44: I couldn't find a lot of information about the genetic architecture of sexually selected traits in the manuscript. Perhaps omit this here and on line 379?

To address this comment, as well as the first comment of referee 1, we have expanded upon the theoretical framework regarding the genetic architecture of sexually selected traits in the results, and added a sentence to the first paragraph of the introduction explaining what we mean by the genetic architecture of fitness.

Comment 35

- L. 111: Not necessarily here, but somewhere in the manuscript there should be a discussion of how good a measure of fitness (lifetime reproductive success) lifetime mating success is. In black grouse, mating success of males seems to be a good measure of genetic parentage, but lifetime reproductive success depends on a number of other life-history traits (clutch size, hatching success, post-hatching survival, etc.) that may not be captured by lifetime mating success. A discussion of potential caveats that arise from this should be included in the paper.

Mating success is consistent with true parentage in black grouse males (see response to comment 26). We agree that lifetime reproductive success depends on both mating success and other traits. However, the most appropriate fitness measure depends on the stage at which sexual selection is assessed – whether at mating, fertilisation (including post-copulatory competition) or offspring survival. In the black grouse system, we believe that pre-copulatory sexual selection (i.e. mating success) is the most relevant measure for the following reasons:

- (i) Female black grouse are monogynous and therefore there is little scope for post-copulatory competition
- (ii) Female clutch size shows very little variation⁸⁻¹²
- (iii) Clutch survival is mainly determined by external factors, such as food availability, temperature and hatching date²³.

Given that we have expanded our manuscript to incorporate the feedback and suggestions of the referees for additional analyses, we have included this caveat in the methods section (lines 475-479). However, we would be happy to write a dedicated discussion paragraph on this topic if the referee and / or the editor deem it important.

Comment 36

- L. 160-162: That total GERP and SnpEff loads are not strongly correlated ($r=0.13$) might seem surprising at first sight. Does this imply that GERP and SnpEff measure complementary aspects of the deleteriousness of mutations, in light of the fact that both loads reduced lifetime mating success?

Thank you for this suggestion. We agree that the lack of a relationship between the total GERP load and the total SnpEff load might seem surprising at first. However, this is not entirely unexpected given that the two approaches rely on different assumptions. Moreover, a previous study also found no relationship between the GERP and SnpEff

loads²⁴. We conclude that these measures capture different aspects of the same underlying phenomenon—the individual mutation load—which explains why both of them are negatively associated with LMS. We have explored this topic further in response to comment 41.

Comment 37

Also, I was wondering whether at the SNP level (rather than the load level) GERP and SnpEff scores are related? To that end it would be interesting to add a panel to Figure 1 which shows some summary statistics (the average?) of the GERP scores for each of the four categories of SnpEff.

Thank you for this suggestion. We found this difficult to integrate into the main figure, as this already contains multiple panels. We therefore added the distribution of GERP scores (including mean values) for SnpEff category in a new extended data figure (Extended Data Figure 1). This shows that GERP and SnpEff also do not produce concordant results at the level of individual mutations.

Comment 38

- L. 164-168, Bayesian GLMMs: A few aspects of these models did not become clear to me. First, I missed an exploration of the sensitivity of these models to the choice of prior, an important element of all Bayesian analyses.

We agree that prior sensitivity analyses are an important check for Bayesian models. To address this criticism, we therefore reran all of our models with two alternative priors: the default brms priors and with an alternative prior specification (population-level effects: mean = 10, SD = 10; intercept: mean = 30, SD = 10). The median beta estimates were all similar and none of our conclusions were altered. We have added more details about this prior sensitivity analysis to the methods (lines 708-712).

Comment 39

Second, I missed the complete model results, including estimates of all the fixed and random effects in the models. Please provide these in the supplementary material. (These first two points also apply to the other Bayesian models in the manuscript).

Thank you for pointing this out. We did not output the full model results as supplementary tables because each LMS model estimates ten or more parameters. Including all of the model outputs would require 39 additional tables. To strike an optimal balance, we have now included full model outputs for the three most important models (i.e. GLMMs of LMS fitting the total GERP load, the total SnpEff load and F_{ROH} as predictors). These outputs can be found in Supplementary Table 2.

Given space limitations, we hope that it is acceptable to refer the reader to our github repository for the full outputs of all of the models, which is now specified in the methods (lines 715-718). However, if necessary, we could include all of the full model outputs in the supplementary materials, although this would significantly increase its length.

For the models based on repeated measures (used for the mediation analysis), we did not include the full model outputs for the same reason, especially as these outputs are even more complex. To illustrate this point, the models of AMS include the mutation load as a fixed effect, together with ID nested within lekking site as a random effect. The R package brms outputs an estimate of the intercept for each of the IDs nested within lek, resulting in a total of 215 parameter estimates per model. With a total of 14 models (one for AMS and one for each of the six sexual traits, separately for both GERP and SnpEff) a full output would exceed 3000 parameter estimates. To keep the results manageable, we have therefore included the model outputs for the fixed effects in Supplementary Tables 9-11, and only report the estimates of the random effects in our github repository (under output/models/intervals).

Comment 40

Third, I was wondering whether a random effect for year-of-birth of the males should be included in the models here, as was done for AMS? In the datasets that I am familiar with, fitness measures vary greatly over time because of environmental effects. If one doesn't account for such environmental effects, differences in fitness become difficult to interpret.

We are indeed aware that environmental effects could potentially complicate the interpretation of fitness measures in heterogenous systems. This is why we incorporated environmental variance into our models by including lekking site as a random effect in the LMS models and sampling year and lekking site as random effects in the AMS models.

We did not include the year of birth of the males in our models because this is only known for the core-males. Therefore, including birth year would require excluding part of our dataset. To show that including birth year does not affect our main conclusions, we re-ran the models using only the core-males (see Table 3 below). The median beta estimates are slightly lower, particularly for total SnpEff load, but the main conclusions remain unchanged. This suggest that temporal variance does not bias our interpretations about the effects of the total load on fitness.

Load	Median beta	95% CI	80% CI	Marginal r^2	Conditional r^2
Total GERP	-0.26	-0.34, -0.17	-0.31, -0.20	0.02	0.08
Total SnpEff	-0.24	-0.33, -0.15	-0.30, -0.18	0.01	0.08

Table 3. Point estimates and credible intervals (CI) of the total GERP load and the total SnpEff load based only on core males, with an additional random effect of birth year.

Comment 41

Lastly, I was curious why the authors ran separate models for GERP and SnpEff loads. Given that they are not strongly correlated and that they might be measuring complementary aspects of load, would it not make sense to include them both in the same model?

Indeed, our results suggest that the total GERP load and the total SnpEff load measure complementary aspects of the mutation load. This is because (i) GERP++ and SnpEff identify different sets of mutations; (ii) the total GERP load and the total SnpEff load are uncorrelated; and (iii) both load measures independently explain variation in LMS.

Nevertheless, as suggested by the reviewer, we fitted the total GERP load and the total SnpEff load together in a single GLMM of LMS. As shown in Fig. 3 below, we found that the effect sizes of both the total GERP load and the total SnpEff load were similar in this combined model (median β estimate of the total GERP load = -0.19, 95% CI = -0.26, -0.13; median β estimate of the total SnpEff load = -0.08, 95% = -0.16, -0.01) compared to the separate models (median β estimate of the total GERP load = -0.21, 95% CI = -0.27, -0.14; median β estimate of the total SnpEff load = -0.11, 95% = -0.18, -0.04). This is exactly what would be expected if their effects are independent.

We decided to model the effect of the total GERP load and the total SnpEff load separately for the following reasons:

- (i) This makes our results easier to compare with previous studies based on one or the other approach.
- (ii) One of our aims was to compare the performance of the two prediction methods. We believe that the relative performance of GERP and SnpEff will be of interest to researchers who do not have access to a genome annotation for their focal species.
- (iii) The two genetic predictors should also be modelled separately due to their contrasting effects (e.g. when looking across the different genomic regions), which is to be expected given the two methods identify mutations based on different assumptions.

Fig. 3. The inferred effects of the total GERP and the total SnpEff load on lifetime mating success using separate models (left) versus a combined model (right). The separate models fitted either the total GERP load or the total SnpEff load as predictor variables, as described in the main text of the manuscript, while the combined model fitted the total GERP load and the total SnpEff load together as predictor variables in a single model. Shown are the posterior distributions of the standardised β estimates of the total GERP load and the total SnpEff load for the separate models (left, identical to Fig. 3a) and the standardised β estimates of the total GERP load and the total SnpEff for the combined model (right). The white circles represent the median posterior estimates, the thick black lines the 80% CIs and the thin black lines the 95% CIs. As the figure indicates, the estimates for the effects of the total GERP load and the total SnpEff load are virtually identical when comparing the separate and combined models.

Comment 42

- L. 182-185: True but the confidence intervals overlap greatly, so that it is difficult to know how much of the effect is due to the GERP score as opposed to stochastic sampling. An important extension here would be to run the models with all loci with GERP score < 0 . This should show a clear and significant difference to the analyses with GERP score ≥ 4 .

We agree that intuitively, a better comparison would be to include models with all SNPs with GERP scores below zero. However, after more carefully reading Huber *et al.*²², which we mentioned in our response to comment 25, it has become clear to us that the GERP scores are not directly proportional to the deleteriousness of a given mutation. This is because GERP does not perform well in distinguishing moderately from strongly

deleterious mutations. Therefore, we have excluded the analysis that calculated total load based on mutations with GERP score 3-4.

Comment 43

- L. 209: *Empirical data also support this expectation, see e.g. Di & Lohmueller 2024, <https://doi.org/10.1093/gbe/evae147>.*

Thank you for pointing us to this interesting reference. We have included one of the references cited in this review and removed the word “theoretical” from this sentence.

Comment 44

- L. 267: *The text suggests that for SnpEff 1000 mutations were randomly sampled, while Extended Data Figure 3 mentions 5000. Please clarify.*

We have now clarified the number of mutations that we randomly sampled in the legend of Supplementary Figure 3.

Comment 45

- L. 297-299: *It would be good to also consider effect sizes here and not only the overlap of the credible intervals with zero. The effect size of SnpEff load on annual mating success (-0.11) was identical to that on lifetime mating success (-0.11, Suppl. Table 1)). Thus, for SnpEff load, the estimated impact on AMS and LMS is identical.*

We agree that conclusions should not be based on a single summary statistic or parameter estimate alone. However, we are cautious about drawing conclusions based on the comparison of the effect sizes of the total SnpEff load between the AMS and LMS models. This is because the two models are very different in their structure: the AMS model uses repeated observations of the same individuals, includes age category as a fixed effect, and adjusts for ID nested within lekking site by fitting this as a random effect. The LMS model does not include repeated measures, includes core vs non-core male as a fixed effect, and adjusts for lekking site by fitting this as a random effect. Additionally, the corresponding 95% CIs for these estimates are large, indicating appreciable uncertainty in the parameter estimates. For these reasons, we do not feel confident in drawing conclusions based on a comparison of the median beta coefficients of the total SnpEff load between the AMS and LMS models.

Comment 46

- L. 337-345: *You could mention here that the importance of heterozygotes was predicted as early as Morton, Crow & Muller 1956 (p. 860: "... most elimination of deleterious genes is now in heterozygotes rather than in homozygotes.)*

Thank you for making this suggestion; We have now cited the reference (line 403-404).

Comment 47

- L. 357: *Given that only GERP load but not SnpEff load affected lek attendance, I think "compelling" is too strong a word here. I'd say that the GERP load results suggest a*

possible pathway by which deleterious mutations affect mating success, but if the case were clear, SnpEff load would have had to yield similar results.

We agree and have toned this statement down accordingly, while also making it clear that it refers only to the total GERP load.

Comment 48

- L. 381: Measures of genetic and genomic load contain little information about population fitness. See Kardos et al. (<https://doi.org/10.1111/mec.17608>) for a recent review.

We have removed the 'population fitness' part from this sentence, and only refer to individual fitness.

Comment 49

- L. 427-430: Please explain what protocol was used for the library preparations and whether it included a PCR step or not.

Thank you for pointing out the missing details in our description of the library preparation methods. We have now added the relevant information to the Supplementary methods ("Library preparation").

Comment 50

- L. 433: 0.1% of a read (150bp) is less than a single base. Is this a typo?

This was not a typo, but a conservative filtering step. We have edited the text to indicate that an entire read was discarded if it contained any N content.

Comment 51

- L. 444: What does "coverage" mean here? Average sequencing coverage was 32, so "coverage" must have a different meaning here.

Thank you for spotting this mistake. We have corrected what the -C flag means in this case in lines 511-512.

Comment 52

- L. 481-482: Why was the Z but not the W chromosome removed from the analyses?

Black grouse have a ZW sex determination system, with males being the homomorphic sex (ZZ). Therefore, in our sampled population of black grouse males, we do not have any W chromosomes.

Comment 53

- Figure 1: I struggled with the colour scheme of Figure 1c because I couldn't differentiate the colours very well. Please use a colour palette that differs more (remember that 8% of men are colour blind...). Also, please add the SFS of neutral SNP

to panels d and e. That way readers can see the difference in SFS between neutral and deleterious loci. Finally, note the typo in the legend of panel f ("Homozygoys").

Thank you for these suggestions. We have updated the figure accordingly, changing the colour scheme and adding the allele frequency distributions of neutral loci (Figure 2).

Comment 54

- Extended Data Figure 1 and Supplementary Table 1: A striking result here is that homozygous and heterozygous GERP load each have a stronger association with LMS (beta is more negative by more than a factor 2) than total GERP load. This is not what one would expect if homozygous and heterozygous loads measure different aspects of the load. Is this result partly explained by a negative correlation between homozygous and heterozygous GERP load, so that the variation in total load is not the sum of the variation in homozygous and heterozygous load? This peculiar result should be explored in more detail.

Indeed, this is a striking result of our analysis. After adding Extended Data Figure 2c, we now show that the homozygous and heterozygous loads are negatively correlated due to their opposing relationships with genomic inbreeding. To understand the contributions of these different load components to fitness, we therefore require a model setup that can estimate their independent effects on fitness, as we have done in our analysis. We have explained this in Extended Data Fig. 4 and elaborated on this finding while referring to the opposing correlations of the homozygous and heterozygous loads with genomic inbreeding (Extended Data Fig. 2c) in lines 252-257 of the Results.

Reviewer #3 (Remarks on code availability):

I reviewed the code but I did not install and run it myself.

The code was very clean and easy to follow. The readme files are clear and contain enough information so that I could find all the relevant pieces of code quickly.

Thank you.

References

1. Kardos, M., Keller, L. F. & Funk, W. C. What Can Genome Sequence Data Reveal About Population Viability? *Molecular Ecology* e17608 (2024)
doi:10.1111/mec.17608.
2. Corrales, C., Pavlovska, M. & Höglund, J. Phylogeography and subspecies status of Black Grouse. *J Ornithol* **155**, 13–25 (2014).
3. Kervinen, M., Alatalo, R. V., Lebigre, C., Siitari, H. & Soulsbury, C. D. Determinants of yearling male lekking effort and mating success in black grouse (*Tetrao tetrix*). *Behavioral Ecology* **23**, 1209–1217 (2012).
4. Burton, P. R., Gurrin, L. C. & Campbell, M. J. Clinical significance not statistical significance: a simple Bayesian alternative to p values. *Journal of Epidemiology & Community Health* **52**, 318–323 (1998).
5. E Alifieris, C., Souferi Chronopoulou, E., T Trafalis, D. & Arvelakis, A. The arbitrary magic of $p < 0.05$: Beyond statistics. *J BUON* **25**, 588–593 (2020).
6. Gelman, A., Hill, J. & Yajima, M. Why We (Usually) Don't Have to Worry About Multiple Comparisons. *Journal of Research on Educational Effectiveness* **5**, 189–211 (2012).
7. Hespanhol, L., Vallio, C. S., Costa, L. M. & Saragiotto, B. T. Understanding and interpreting confidence and credible intervals around effect estimates. *Brazilian Journal of Physical Therapy* **23**, 290–301 (2019).
8. McElreath, R. *Statistical Rethinking: A Bayesian Course with Examples in R and Stan*. (Chapman and Hall/CRC, 2018). doi:10.1201/9781315372495.

9. Dussex, N., Morales, H. E., Grossen, C., Dalén, L. & Van Oosterhout, C. Purging and accumulation of genetic load in conservation. *Trends in Ecology & Evolution* S0169534723001313 (2023) doi:10.1016/j.tree.2023.05.008.
10. Huber, C. D., Kim, B. Y. & Lohmueller, K. E. Population genetic models of GERP scores suggest pervasive turnover of constrained sites across mammalian evolution. *PLoS Genet* **16**, e1008827 (2020).
11. Hong, E. P. & Park, J. W. Sample Size and Statistical Power Calculation in Genetic Association Studies. *Genomics Inform* **10**, 117 (2012).
12. Stoffel, M. A., Johnston, S. E., Pilkington, J. G. & Pemberton, J. M. Genetic architecture and lifetime dynamics of inbreeding depression in a wild mammal. *Nat Commun* **12**, 2972 (2021).
13. Bertorelle, G. *et al.* Genetic load: genomic estimates and applications in non-model animals. *Nat Rev Genet* **23**, 492–503 (2022).
14. Herdegen, M. & Radwan, J. Effect of induced mutations on sexually selected traits in the guppy, *Poecilia reticulata*. *Animal Behaviour* **110**, 105–111 (2015).
15. Almbro, M. & Simmons, L. W. Sexual selection can remove an experimentally induced mutation load: brief communication. *Evolution* **68**, 295–300 (2014).
16. Lebigre, C., Alatalo, R. V., Siitari, H. & Parri, S. Restrictive mating by females on black grouse leks. *Molecular Ecology* **16**, 4380–4389 (2007).
17. von Seth, J. *et al.* Genomic trajectories of a near-extinction event in the Chatham Island black robin. *BMC Genomics* **23**, 747 (2022).
18. Hasselgren, M. *et al.* Strongly deleterious mutations influence reproductive output and longevity in an endangered population. *Nat Commun* **15**, 8378 (2024).

19. Davydov, E. V. *et al.* Identifying a High Fraction of the Human Genome to be under Selective Constraint Using GERP++. *PLoS Comput Biol* **6**, e1001025 (2010).
20. Smeds, L. & Ellegren, H. From high masked to high realized genetic load in inbred Scandinavian wolves. *Molecular Ecology* **32**, 1567–1580 (2023).
21. Bortoluzzi, C. *et al.* Lepidoptera genomics based on 88 chromosomal reference sequences informs population genetic parameters for conservation. Preprint at <https://doi.org/10.1101/2023.04.14.536868> (2023).
22. Huber, C. D., Kim, B. Y. & Lohmueller, K. E. Population genetic models of GERP scores suggest pervasive turnover of constrained sites across mammalian evolution. *PLoS Genet* **16**, e1008827 (2020).
23. Ludwig, G. X., Alatalo, R. V., Helle, P. & Siitari, H. Individual and Environmental Determinants of Daily Black Grouse Nest Survival Rates at Variable Predator Densities. *Annales Zoologici Fennici* **47**, 387–397 (2010).
24. Smeds, L. & Ellegren, H. From high masked to high realized genetic load in inbred Scandinavian wolves. *Molecular Ecology* **32**, 1567–1580 (2023).

Dear editor and reviewers,

Thank you for your additional feedback on our revised manuscript. We have made further changes to incorporate the comments and suggestions made by all three reviewers. Please see below a detailed response to each of the reviewer's comments.

Reviewers' comments:

Reviewer #1 (Remarks to the Author):

I would like to thank the authors of doing a good job addressing the comments by all reviewers and clarifying most of our concerns and still consider this a highly interesting and relevant paper. I just have a few smaller comments:

Thank you for the positive feedback on our revised manuscript.

Comment 1

The F_{ROH} estimates are - as far as I can understand based on all sizes of ROH. I still find the demographic history of the study population a bit unclear, did you look into the distribution of recent vs more ancient inbreeding signatures? Since this can influence the distribution of mutational load in the population, this might be an aspect worth considering.

Yes, the F_{ROH} estimates were based on all sizes of ROH. To clarify the demographic history in the context of inbreeding, we have now elaborated on the ROH length distribution and have converted the ROH lengths into time in generations and years ago. We show that short ROHs constitute the majority of all ROHs and contribute most to F_{ROH} . This indicates that inbreeding in our focal population of black grouse is mostly historical (>50 generations ago, equivalent to approx. >150 years ago). We have expanded on the ROH length distribution in lines 150-159 of the Results and in lines 591-597 of the Methods.

Comment 2

I am a bit puzzled by the use of lifetime mating success as being directly linked to fitness. As I understand, this is based on the total number of observed matings. To what extent has it been verified that LMS is correlated to LRS, i.e. can matings occur without any chicks being produced? I totally agree about this being an important trait, but I feel like the link to fitness (survival and reproduction) could benefit from further clarification.

Thank you for this comment. We agree that clarifying the link between lifetime mating success (LMS) and fitness is important. LMS is strongly correlated to LRS for the following reasons:

- (i) In black grouse, both multiple matings and multiple paternities are rare, with females typically mating only once with a single male to produce the entire clutch¹.
- (ii) Infertile clutches are uncommon (<1%), meaning that copulations almost always result in successful fertilization, with the male siring the entire clutch¹.

While LRS may be influenced by additional factors such as clutch size, hatching success and chick survival, these are predominantly influenced by environmental variation² and not by male quality. Thus, we believe that lifetime mating success is both the most direct and appropriate measure of male fitness in this species.

To incorporate the above points, we have now expanded on the use of lifetime mating success as an indicator of fitness in lines 521-528 of the Methods.

Comment 3

Has inbreeding avoidance been documented in this species?

There is no evidence that black grouse actively avoid inbreeding, but they do avoid it passively through female-biased dispersal³ and because males and females have different reproductive lifespans, minimising the overlap of potentially related individuals from a given cohort⁴. We have added this information to lines 144-147 of the Results.

Comment 4

The discrepancies between the GERP and SnpEff has been adressed by the authors, but I would like the authors to motivate the inclusion of both estimates given the strong divergence in the number of mutations identified.

Indeed, GERP identified many more deleterious mutations ($n = 413,489$) than SnpEff did ($n = 5,341$). This difference is not unexpected as the two methods rely on different assumptions and search for mutations across different sections of the genome (SnpEff is limited to genes). Nevertheless, it is common that studies only use one of the two methods to predict deleterious mutations (e.g. ^{5,6}). We therefore wanted to test whether using these prediction methods would lead to the same conclusions. Including both approaches in our manuscript provides a more complete story both from a methodological and a biological perspective. Our results indicate that using these different methods could potentially result in different conclusions being reached and/or different levels of confidence.

We have now specified why we included both methods in lines 162-164 of the Results and elaborate on the discrepancy between the absolute number of deleterious mutations identified by the two methods in lines 189-198 in the Results.

Reviewer #2 (Remarks to the Author):

The authors have done great work addressing the comments and have significantly improved the manuscript.

Thank you for this positive feedback.

Comment 5

I only have one major concern regarding the GO analysis. The authors selected six GO terms and only did analyses for genes belonging to these GO terms. I'm afraid that it is

not an appropriate way to “hand pick” GO terms for functional analyses. I agree that the biological functions of these GO terms are theoretically important, but so as many others. For example, why choose “androgen metabolic process”, but not the parent term “hormone metabolic process”, as hormone metabolism in general is associated with behavior, body growth, immunity level and so on. My point here is that a functional enrichment analysis is more appropriate to find key functional terms, as reproduction is a complex process, expected to be associated with many major functional pathways. I’d expect genes involved in biological processes important for reproductive success are less likely to harbor deleterious alleles and thus should be under-represented. Besides R packages like topGO, there are also online tools for GO enrichment analysis which might be handy, for example: <https://biit.cs.ut.ee/gprofiler/gost>

Thank you for this comment. We respectfully disagree with the reviewer’s concern regarding our GO analysis approach. The analysis we conducted is conceptually similar to a candidate gene approach, a widely used method in genetics (particularly clinical genetics)⁷. In this approach, hypotheses are formulated based on existing knowledge, which then guide the selection of specific candidate biological processes for investigation. In our case, we selected six GO terms based on decades of research on black grouse and sexual selection in general. Specifically, we focused on the roles of androgens (including testosterone which strongly correlates with male black grouse sexual trait expression^{8,9}), cellular respiration (crucial for energetically costly behaviours like lekking¹⁰), developmental growth (essential for body growth, which predicts survival² and territoriality in black grouse¹⁰), immunity (hypothesized for decades to link testosterone-dependent sexual traits to honest signaling^{11,12}), muscle tissue development (greater glycogen storage in muscles allows higher display rates in black grouse males^{13,14}) and oxidative stress (which can affect carotenoid and melanin-based ornamental traits¹⁵).

Our hypotheses and rationale are clearly documented (with literature references) in Supplementary table 5.

We formulated these hypotheses to test whether deleterious mutations in genes associated with these key processes negatively affect lifetime mating success (LMS) in black grouse. While we acknowledge that there may be other relevant biological processes, the GO terms we selected were driven by strong, *a priori*, hypotheses rather than being “hand-picked.” We would like to note that none of the other reviewers raised concerns about our GO term selection.

To better clarify our approach, we have included additional information in the manuscript (lines 352-363 of the Results) and explicitly stated in lines 369-371 that our analysis does not cover all biological processes potentially relevant to LMS.

Regarding the reviewer’s suggestion of conducting a functional enrichment analysis, we are unclear about the logic behind the expectation that mutations in genes important for reproductive success should be underrepresented. We assume the reviewer means that these genes are less likely to harbor deleterious mutations because these mutations will already have been purged and should therefore be underrepresented in

the adult population. However, our main results show that the total mutation load is negatively associated with lifetime mating success, which indicates that deleterious mutations do in fact persist in the adult population, where they reduce reproductive success. If our interpretation of the reviewer’s expectation is correct, this would not align with our findings.

Additionally, the suggested GO enrichment analysis would assess which GO terms are overrepresented for deleterious mutations. However, this is the opposite of the question formulated by the reviewer (“I’d expect genes involved in biological processes important for reproductive success are *less* likely to harbor deleterious alleles and thus should be *under*-represented”). This is because a GO enrichment analysis does not tell you which GO terms are *under*represented, but rather which ones are *over*represented.

Furthermore, for the GERP analysis, the suggested approach in essence answers which GO terms are more evolutionarily conserved than others. However, the GERP approach inherently assumes that conserved genes play key roles in processes related to survival and reproduction, making the proposed analysis somewhat circular.

The proposed analysis also overlooks individual variation in both the mutation load and fitness, which is a central aspect of our study. Our individual-based approach allows us to investigate the relationship between mutation load and fitness more directly.

Nevertheless, we implemented the proposed analysis to identify GO terms that are enriched for deleterious mutations. First, we extracted genes containing predicted deleterious mutations separately for GERP and SnpEff. We then compared these unranked lists of target gene IDs to unranked lists of background genes containing at least one SNP, regardless of their deleteriousness. We implemented this analysis in GOrilla¹⁶ using the default settings. While we found no significant enrichment for SnpEff mutations, several GO terms were identified that were enriched for GERP mutations. These significant GO terms are mostly associated with generic biological processes such as binding and catalytic activity (see Table 1 below), which are either highly evolutionarily conserved, or which might in some cases might represent false positives, as predicted deleterious mutations associated with these terms are still abundant in the population and hence appear not to be purged. Given uncertainty over whether these mutations are truly deleterious, we do not believe that the results of this enrichment analysis contribute significantly to the manuscript.

We hope this explanation clarifies our reasoning and approach. Thanks again for the thoughtful feedback.

Table 1. Significant GO terms that are enriched for deleterious GERP mutations.

GO Term	Description	FDR q-value	Enrichment	B	b
GO:0003824	catalytic activity	9.39 e ⁻⁴	1.03	3513	2963
GO:0005524	ATP binding	6.05 e ⁻⁴	1.07	1001	874

GO:0030554	adenyl nucleotide binding	5.26 e ⁻⁴	1.07	1042	908
GO:0032559	adenyl ribonucleotide binding	4.14 e ⁻⁴	1.07	1035	902
GO:0046872	metal ion binding	1.10 e ⁻³	1.04	2333	1982
GO:0043169	cation binding	1.27 e ⁻³	1.04	2385	2024
GO:0008144	drug binding	1.32 e ⁻³	1.06	1130	978
GO:0043167	ion binding	1.62 e ⁻³	1.03	3673	3085
GO:0140096	catalytic activity, acting on a protein	2.74 e ⁻³	1.05	1342	1153
GO:0032553	ribonucleotide binding	4.03 e ⁻²	1.05	1245	1064
GO:0043168	anion binding	4.50 e ⁻²	1.04	1862	1576
GO:0000166	nucleotide binding	4.91 e ⁻²	1.04	1411	1201
GO:0017076	purine nucleotide binding	4.76 e ⁻²	1.05	1243	1061

Here are several minor comments and replies to the replies by the order of line numbers:

Comment 6

Line 47-48 (comment 18): I think the current expression is clear and precise. However the content in the brackets “(the reduction in fitness due to deleterious mutations)” is repetitive and can be deleted.

We see why a reader might perceive this sentence as being repetitive. However as this is the first time we mention the term “mutation load”, we need to define it here. We have now edited the definition slightly to avoid too much repetition in lines 47-48.

Comment 7

Line 49 (comment 19): Thanks for the kind explanation, but what I didn’t understand here is that, why it is mentioned here as the effect of de novo (new) mutations is not part of this study. Maybe the authors are trying to refer here is that sexual traits can be affected by deleterious mutations? Maybe consider rephrasing in a way, for example “... and experimental studies showed that sexual traits can be disrupted by deleterious mutations. However, key attributes ... remain poorly understood in wild populations”

Thank you for the helpful suggestion. The sentence the reviewer refers to refers to mutation induction experiments. These experiments have shown that experimentally induced mutations disrupt sexual trait expression and reduce fitness. By definition, these induced mutations are *de novo*. However, note that not all induced *de novo* mutations are deleterious: they can be neutral, beneficial, or deleterious¹⁷. Consequently, we prefer not to adopt the suggested rephrasing, as this runs the risk of implying that all induced mutations are deleterious, which would be incorrect.

Comment 8

Line 85-87: I’d suggest being cautious to state that estimating genetic load is a better way than estimating inbreeding. There’s still not a good way to really estimate fitness effects of the “deleterious mutations” in wild populations. And there are studies

showing correlations between fitness and inbreeding (e.g. DOI: 10.1016/j.cub.2019.06.064). What is the best way to assess fitness is still very much under debate.

We agree that this statement requires caution. Indeed, whether the genetic load explains more variation in fitness than inbreeding depends on the accuracy of both deleterious mutation prediction and inbreeding estimation. We have now clarified on lines 83-89 that this expectation holds in theory, but whether it holds in practice depends on the precision of these estimates.

Comment 9

Line 132 (comment 7): I think what Reviewer 1 meant is the coverage (depth) range for the individuals, rather than for the SNPs. It is necessary to show that no sample with low depth is included because for example, for an individual with an average depth of 8x, a minimum SNP depth of 20 would only keep SNPs in regions with depth higher than twice of the genome-wide average, likely repeated regions.

Thank you for pointing this out. We have now additionally calculated the mean coverage across SNPs per individual, which has a range of 21.62–32.58X. We have included this information in the same paragraph on lines 563-564. We believe these values are within the expected ranges and therefore did not exclude any individuals or additional SNPs based on this information.

Comment 10

Line 136 (comment 24): The number of related pairs does not reflect the number of individuals involved. Again I don't think relatedness in the samples affects the main results, but I'd suggest providing additional information on how many of the 190 individuals are found to be related to each other, rather than only a slightly misleading number of the proportion of the pairs.

We appreciate the reviewer's suggestion to add more information about the relatedness structure of our dataset. Relatedness is inherently a pairwise measure, which is why we reported the numbers of related pairs. To complement this information, we now additionally summarise the number of individuals with at least one relative in the dataset. Specifically, we now state: "Out of the 190 individuals, 104 had at least one first-degree relative (full sibling or parent-offspring), 124 had at least one second-degree relative, and 143 had at least one third-degree relative." This information has been added to lines 33–36 of the Supplementary Information to improve clarity. We hope this addresses the reviewer's comment.

Comment 11

Line 200 (comment 13): Thank you very much for the detailed explanation. I agree that the Bayesian framework applied in this study is appropriate. What I meant initially was to provide p values converted from the standardized beta just to make the results easier

to follow, but you did make a point that providing the standardized beta is still the best way. I'd suggest having a very brief statement on how to interpret the results somewhere in this session or in the Methods, just to guide readers like me. For example, from your response: "we consider an effect to be "statistically significant" if its 95% CI does not overlap zero"

Thank you for the clarification about your initial comment. We have now included a statement as proposed by the reviewer in lines 775-777.

Comment 12

Line 301-312 (comment 14): Thanks for sharing the preliminary results. I assume these are from ongoing works. This is very exciting and looking forward to your further studies. However, stating there are unpublished results without giving proof is odd. I think it is enough to mention that one possible explanation is that highly deleterious mutations are selected at the early stage of life, and further study is needed.

Indeed, these are results are from ongoing research. At the risk of providing a less compelling answer to the previous comment made by the same reviewer, we have now excluded the sentence where we state that preliminary data support the explanation that exonic GERP mutations are purged during early life. The change can be seen in lines 339-349.

Comment 13

Line 802: There is a "standardized" while the rest are "standardised".

Thank you, this has now been corrected.

Comment 14

Reviewer #2 (Remarks on code availability):

The codes are comprehensive and very well organized. The README is very clear and it is very easy to locate scripts for every analysis. I have not further checked the codes in this round of review.

Thank you.

Reviewer #3 (Remarks to the Author):

Genetic architecture of male reproductive success in a lekking bird: insights from predicted deleterious mutations

I greatly appreciate the effort the authors have put into the revision of their manuscript, and into explaining their decisions in detail. To my mind, the result is a very nice contribution to the field.

Thank you for your positive feedback on the revisions.

Comment 15

There is only one issue that I think the authors have not dealt with convincingly. It concerns the very large number of SNPs that are categorised as deleterious by both GERP and SnpEff (comment 28 in the rebuttal letter). Here is why I think the issue should be addressed head on in the manuscript:

I agree that comparing the absolute number of deleterious SNPs is not straightforward for a number of reasons which the authors highlight in their rebuttal letter. Nevertheless, the absolute number of deleterious alleles is what determines the genetic load (lines 614-616 in the revised manuscript), and hence the absolute number of deleterious SNPs is an essential quantity that cannot be ignored.

The authors argue that other papers have also found such high numbers of deleterious SNPs. While one study of wolves has done so, some studies of birds have reported a much smaller number. For example, Mather et al. (2023, Evolution) have found ~12'700 deleterious SNPs among 66 sequenced Montezuma Quails with a total of 9.3 Mio. SNPs. This corresponds to a percentage of deleterious SNPs of approx. 0.14%. Even accounting for the differences in sample size, this is nearly ten times less than what the current paper reports.

To my mind, the value of the present study is undisputed. The very high number of predicted deleterious alleles does not distract from that. It simply highlights the fact that it still isn't straightforward to identify deleterious SNPs with a high level of confidence, especially in non-model organisms where annotation and mapping errors are normal. To my mind, all that is needed is a short paragraph highlighting the comparatively high number of deleterious SNPs discovered in this study, and a brief exploration of the possible reasons for it. Annotation and sequencing errors (see reference 49 in the manuscript) would be obvious candidates, but some of the reasons given by the authors in their rebuttal letter might also contribute. All in all, a short paragraph highlighting the difficulty of identifying deleterious SNPs will strengthen and not weaken the impact of this manuscript.

We agree that it is both interesting and important to consider why the absolute number of deleterious mutations varies so markedly across study systems and with the methodological approaches used. Thank you also for providing another example against which we can compare the empirical proportion of deleterious mutations in the black grouse. We would like to highlight that the cited study further underscores the challenges of comparing absolute mutation counts across populations while also highlighting the influence of demographic history. Specifically, this study reports differences in the mutation load of two populations, variation over time relative to a bottleneck, and a small discrepancy between simulated and empirical estimates. We have now included this reference in our manuscript to provide additional context.

As suggested, we have now added a new paragraph to the Results covering this topic, which can be found on lines 200-212. We use this to highlight differences in the

absolute number of deleterious mutations across studies and touch upon a number of potential explanations for these differences. In response to comment 4 in the current rebuttal, we also touch upon relevant methodological aspects on lines 189-198.

References

1. Lebigre, C., Alatalo, R. V., Siitari, H. & Parri, S. Restrictive mating by females on black grouse leks. *Molecular Ecology* **16**, 4380–4389 (2007).
2. Ludwig, G. X., Alatalo, R. V., Helle, P. & Siitari, H. Individual and Environmental Determinants of Daily Black Grouse Nest Survival Rates at Variable Predator Densities. *Annales Zoologici Fennici* **47**, 387–397 (2010).
3. Lebigre, C., Alatalo, R. V. & Siitari, H. Female-biased dispersal alone can reduce the occurrence of inbreeding in black grouse (*Tetrao tetrix*). *Molecular Ecology* **19**, 1929–1939 (2010).
4. Soulsbury, C. D., Alatalo, R. V., Lebigre, C. & Siitari, H. Restrictive mate choice criteria cause age-specific inbreeding in female black grouse, *Tetrao tetrix*. *Animal Behaviour* **83**, 1497–1503 (2012).
5. Dussex, N. *et al.* Constraints to gene flow increase the risk of genome erosion in the Ngorongoro Crater lion population. *Commun Biol* **8**, 640 (2025).
6. Hasselgren, M. *et al.* Strongly deleterious mutations influence reproductive output and longevity in an endangered population. *Nat Commun* **15**, 8378 (2024).
7. Kwon, J. M. & Goate, A. M. The candidate gene approach. *Alcohol Res Health* **24**, 164–168 (2000).
8. Rintamaki, P. T. Combs and sexual selection in black grouse (*Tetrao tetrix*). *Behavioral Ecology* **11**, 465–471 (2000).
9. Testosterone and male mating success on the black grouse leks. *Proc. R. Soc. Lond. B* **263**, 1697–1702 (1996).
10. Kervinen, M., Alatalo, R. V., Lebigre, C., Siitari, H. & Soulsbury, C. D. Determinants of yearling male lekking effort and mating success in black grouse (*Tetrao tetrix*). *Behavioral Ecology* **23**, 1209–1217 (2012).
11. Zahavi, A. & Zahavi, A. *The Handicap Principle: A Missing Piece of Darwin's Puzzle*. (Oxford University Press, 1999).
12. Kurtz, J. & Sauer, K. P. The immunocompetence handicap hypothesis: testing the genetic predictions. *Proc. R. Soc. Lond. B* **266**, 2515–2522 (1999).
13. Briffa, M. & Sneddon, L. U. Physiological constraints on contest behaviour. *Functional Ecology* **21**, 627–637 (2007).
14. Guderley, H. & Couture, P. Stickleback Fights: Why Do Winners Win? Influence of Metabolic and Morphometric Parameters. *Physiological and Biochemical Zoology* **78**, 173–181 (2005).
15. Costantini, D. The Oxidative Costs of a Colourful Life. in *The Role of Organismal Oxidative Stress in the Ecology and Life-History Evolution of Animals* 287–322 (Springer Nature Switzerland, Cham, 2024). doi:10.1007/978-3-031-65183-0_8.

16. Eden, E., Navon, R., Steinfeld, I., Lipson, D. & Yakhini, Z. GOrilla: a tool for discovery and visualization of enriched GO terms in ranked gene lists. *BMC Bioinformatics* **10**, 48 (2009).
17. Bao, K., Melde, R. H. & Sharp, N. P. Are mutations usually deleterious? A perspective on the fitness effects of mutation accumulation. *Evol Ecol* **36**, 753–766 (2022).